# Noise-Contrastive Estimation for Multivariate Point Processes

**Hongyuan Mei**    **Tom Wan**    **Jason Eisner**
Department of Computer Science, Johns Hopkins University
3400 N. Charles Street, Baltimore, MD 21218 U.S.A
{hmei,tom,jason}@cs.jhu.edu

## Abstract

The log-likelihood of a generative model often involves both positive and negative terms. For a temporal multivariate point process, the negative term sums over all the possible event types at each time and also integrates over all the possible times. As a result, maximum likelihood estimation is expensive. We show how to instead apply a version of noise-contrastive estimation—a general parameter estimation method with a less expensive stochastic objective. Our specific instantiation of this general idea works out in an interestingly non-trivial way and has provable guarantees for its optimality, consistency and efficiency. On several synthetic and real-world datasets, our method shows benefits: for the model to achieve the same level of log-likelihood on held-out data, our method needs considerably fewer function evaluations and less wall-clock time.

## 1   Introduction

Maximum likelihood estimation (MLE) is a popular training method for generative models. However, to obtain the likelihood of a generative model given the observed data, one must compute the probability of each observed sample, which often includes an expensive normalizing constant. For example, in a language model, each word is typically drawn from a softmax distribution over a large vocabulary, whose normalizing constant requires a summation over the vocabulary.

This paper aims to alleviate a similar computational cost for multivariate point processes. These generative models are natural tools to analyze streams of discrete events in continuous time. Their likelihood is improved not only by raising the probability of the observed events, but by lowering the probabilities of the events that were observed *not* to occur. There are infinitely many times at which no event of any type occurred; to predict these *non*-occurrences, the likelihood must integrate the infinitesimal event probability for each event type over the entire observed time interval. Therefore, the likelihood is expensive to compute, particularly when there are many possible event types.

As an alternative to MLE, we propose to train the model by learning to discriminate the observed events from events sampled from a noise process. Our method is a version of **noise-contrastive estimation** (NCE), which was originally developed for unnormalized (energy-based) distributions and then extended to conditional softmax distributions such as language models. To our best knowledge, we are the first to extend the method and its theoretical guarantees (for optimality, consistency and efficiency) to the context of multivariate point processes. We will also discuss similar efforts in related areas in section 4.

On several datasets, our method shows compelling results. By evaluating fewer event intensities, training takes much less wall-clock time while still achieving competitive log-likelihood.

## 2 Preliminaries

### 2.1 Event Streams and Multivariate Point Processes

Given a fixed time interval $[0, T)$, we may observe an **event stream** $x_{[0,T)}$: at each continuous time $t$, the observation $x_t$ is one of the discrete types $\{\varnothing, 1, \dots, K\}$ where $\varnothing$ means *no event*. An non-$\varnothing$ observation is called an **event**. A generative model of an event stream is called a **multivariate point process**.[*]

We wish to fit an **autoregressive** probability model to observed event streams. In a discrete-time autoregressive model, events would be generated from left to right, where $x_t$ is drawn from a distribution that depends on $x_0, \dots, x_{t-1}$. The continuous-time version still generates events from left to right,[1] but at any specific time $t$ we have $p(x_t = \varnothing) = 1$, with only an infinitesimal probability of any event. (For a computationally practical sampling method, see section 3.1.) The model is a stochastic process defined by functions $\lambda_k$ that determine a finite **intensity** $\lambda_k(t \mid x_{[0,t)}) \geq 0$ for each event type $k \neq \varnothing$ at each time $t > 0$. This intensity depends on the **history** of events $x_{[0,t)}$ that were drawn at times $< t$. It quantifies the **instantaneous rate** at time $t$ of events of type $k$. That is, $\lambda_k(t \mid x_{[0,t)})$ is the limit as $dt \to^+ 0$ of $\frac{1}{dt}$ times the expected number of events of type $k$ on the interval $[t, t + dt)$, where the expectation is conditioned on the history.

As the event probabilities are infinitesimal, the times of the events are almost surely distinct. To ensure that we have a point process, the intensity functions must be chosen such that the total number of events on any bounded interval is almost surely finite. Models of this form include inhomogeneous Poisson processes (Daley & Vere-Jones, 2007), in which the intensity functions ignore the history, as well as (non-explosive) Hawkes processes (Hawkes, 1971) and their modern neural versions (Du et al., 2016; Mei & Eisner, 2017).

Most models use intensity functions that are continuous between events. Our analysis requires only

**Assumption 1** (Continuity). *For any event stream $x_{[0,T)}$ and event type $k \in \{1, \dots, K\}$, $\lambda_k(t \mid x_{[0,t)})$ is Riemann integrable, i.e., bounded and continuous almost everywhere w.r.t. time $t$.*

### 2.2 Maximum Likelihood Estimation: Usefulness and Difficulties

In practice, we parameterize the intensity functions by $\theta$. We write $p_\theta$ for the resulting probability density over event streams. When learning $\theta$ from data, we make the conventional assumption that the true point process $p^*$ actually falls into the chosen model family:

**Assumption 2** (Existence). *There exists at least one parameter vector $\theta^*$ such that $p_{\theta^*} = p^*$.*

Then as proved in Appendix A, such a $\theta^*$ can be found as an argmax of

$$J_{\mathrm{LL}}(\theta) \overset{\mathrm{def}}{=} \mathbb{E}_{x_{[0,T)} \sim p^*} \left[ \log p_\theta(x_{[0,T)}) \right] \tag{1}$$

Given assumption 1, the $\theta$ values that maximize $J_{\mathrm{LL}}(\theta)$ are exactly the set $\Theta^*$ of values for which $p_\theta = p^*$: any $\theta$ for which $p_\theta \neq p^*$ would end up with a strictly smaller $J_{\mathrm{LL}}(\theta)$ by increasing the cross entropy $-p^* \log p_\theta$ over some interval $(t, t')$ for a set of histories with non-zero measure.

If we modify equation (1) to take the expectation under the empirical distribution of event streams $x_{[0,T)}$ in the training dataset, then $J_{\mathrm{LL}}(\theta)$ is proportional to the log-likelihood of $\theta$. For any $x_{[0,T)}$ that satisfies the condition in assumption 1, the log-density used in equation (1) can be expressed in terms of $\lambda_k(t \mid x_{[0,t)})$:

$$\log p_\theta(x_{[0,T)}) = \sum_{t:x_t \neq \varnothing} \log \lambda_{x_t}(t \mid x_{[0,t)}) - \int_{t=0}^{T} \sum_{k=1}^{K} \lambda_k(t \mid x_{[0,t)}) dt \tag{2}$$

Notice that the second term lacks a log. It is *expensive* to compute in the following cases:

- The total number of event types $K$ is large, making $\sum_{k=1}^{K}$ slow.
- The integral $\int_{t=0}^{T}$ is slow to estimate well, e.g., via a Monte Carlo estimate $\frac{T}{J} \sum_{j=1}^{J} \sum_{k=1}^{K} \lambda_k(t_j)$ where each $t_j$ is randomly sampled from the uniform distribution over $[0, T)$.
- The chosen model architecture makes it hard to parallelize the $\lambda_k(t_j)$ computation over $j$ and $k$.

---

[*]This paper uses endnotes instead of footnotes. They are found at the start of the supplementary material.

## 2.3 Noise-Contrastive Estimation in Discrete Time

For autoregressive models of *discrete-time* sequences, a similar computational inefficiency can be tackled by applying the principle of noise-contrastive estimation (Gutmann & Hyvärinen, 2010), as follows. For each history $x_{0:t} \stackrel{\text{def}}{=} x_0 x_1 \dots x_{t-1}$ in training data, NCE trains the model $p_\theta$ to discriminate the actually observed datum $x_t$ from some noise samples whose distribution $q$ is known. The intuition is: optimal performance is obtained *if and only if* $p_\theta$ matches the true distribution $p^*$.

More precisely, given a bag $\{x_t^0, x_t^1, \dots, x_t^M\}$, where exactly one element of the bag was drawn from $p^*$ and the rest drawn i.i.d. from $q$, consider the log-posterior probability (via Bayes' Theorem[2]) that $x_t^0$ was the one drawn from $p^*$:

$$\log \frac{p^*(x_t^0|x_{0:t}) \prod_{m=1}^M q(x_t^m|x_{0:t})}{\sum_{m=0}^M p^*(x_t^m|x_{0:t}) \prod_{m' \neq m} q(x_t^{m'}|x_{0:t})} \tag{3}$$

The "ranking" variant of NCE (Jozefowicz et al., 2016) substitutes $p_\theta$ for $p^*$ in this expression, and seeks $\theta$ (e.g., by stochastic gradient ascent) to maximize the expectation of the resulting quantity when $x_t^0$ is a random observation in training data,[3] $x_{0:t}$ is its history, and $x_t^1, \dots, x_t^M$ are drawn i.i.d. from $q(\cdot \mid x_{0:t})$.

This objective is really just conditional maximum log-likelihood on a supervised dataset of $(M+1)$-way classification problems. Each problem presents an unordered set of $M + 1$ samples—one drawn from $p^*$ and the others drawn i.i.d. from $q$. The task is to guess *which* sample was drawn from $p^*$. Conditional MLE trains $\theta$ to maximize (in expectation) the log-probability that the model assigns to the correct answer. In the infinite-data limit, it will find $\theta$ (if possible) such that these log-probabilities *match* the true ones given by (3). For that, it is *sufficient* for $\theta$ to be such that $p_\theta = p^*$. Given assumption 2, Ma & Collins (2018) show that $p_\theta = p^*$ is also *necessary*, i.e., the NCE task is sufficient to find the true parameters. Although the NCE objective does not learn to predict the full observed sample $x_t$ as MLE does, but only to distinguish it from the $M$ noise samples, their theorem implies that in expectation over all possible sets of $M$ noise samples, it actually retains all the information (provided that $M > 0$ and $q$ has support everywhere that $p^*$ does).

This NCE objective is computationally cheaper than MLE when the distribution $p_\theta(\cdot \mid x_{0:t})$ is a softmax distribution over $\{1, \dots, K\}$ with large $K$. The reason is that the expensive normalizing constants in the numerator and denominator of equation (3) need not be computed. They cancel out because all the probabilities are conditioned on the same (actually observed) history.

## 3 Applying Noise-Contrastive Estimation in Continuous Time

The expensive $\int \sum$ term in equation (2) is rather similar to a normalizing constant,[4] as it sums over non-occurring events. We might try to avoid computing it[5] by discretizing the time interval $[0, T)$ into finitely many intervals of width $\Delta$ and applying NCE. In this case, we would be distinguishing the true sequence of events on an interval $[i\Delta, (i + 1)\Delta)$ from corresponding noise sequences on the same interval, given the same (actually observed) history $x_{[0, i\Delta)}$. Unfortunately, the distribution $p_\theta(\cdot \mid x_{[0, i\Delta)})$ in the objective still involves an $\int \sum$ term where the integral is over $[i\Delta, (i + 1)\Delta)$ and the inner sum is over $k$. The solution is to shrink the intervals to *infinitesimal width $dt$*. Then our log-posterior over each of them becomes

$$\log \frac{p_\theta(x_{[t,t+dt)}^0 \mid x_{[0,t)}^0) \prod_{m=1}^M q(x_{[t,t+dt)}^0 \mid x_{[0,t)}^0)}{\sum_{m=0}^M p_\theta(x_{[t,t+dt)}^m \mid x_{[0,t)}^0) \prod_{m' \neq m} q(x_{[t,t+dt)}^{m'} \mid x_{[0,t)}^0)} \tag{4}$$

We will define the noise distribution $q$ in terms of finite intensity functions $\lambda_k^q$, like the ones $\lambda_k$ that define $p_\theta$. As a result, at a *given* time $t$, there is only an infinitesimal probability that *any* of $\{x_t^0, x_t^1, \dots, x_t^M\}$ is an event. Nonetheless, at *each* time $t \in [0, T)$, we will consider generating a noise event (for each $m > 0$) conditioned on the actually observed history $x_{[0,t)}$. Among these uncountably many times $t$, we may have some for which $x_t^0 \neq \varnothing$ (the observed events), or where $x_t^m \neq \varnothing$ for some $1 \leq m \leq M$ (the noise events).

Almost surely, the set of times $t$ with a real or noise event remains finite. Our NCE objective is the expected sum of equation (4) over all such times $t$ in an event stream, when the stream is drawn uniformly from the set of streams in the training dataset—as in section 6—and the noise events are then drawn as above.

Our objective ignores all other times $t$, as they provide no information about $\theta$. After all, when $x_t^0 = \cdots = x_t^M = \varnothing$, the probability that $x_t^0$ is the one drawn from the true model must be $1/(M+1)$ by symmetry, regardless of $\theta$. At these times, the ratio in equation (4) does reduce to $1/(M+1)$, since all probabilities are 1.

At the times $t$ that we do consider, how do we compute equation (4)? Almost surely, exactly one of $x_t^0, \ldots, x_t^M$ is an event $k$ for some $k \neq \varnothing$. As a result, exactly one factor in each product is infinitesimal ($dt$ times the $\lambda_k$ or $\lambda_k^{\mathrm{q}}$ intensity), and the other factors are 1. Thus, the $dt$ factors cancel out between numerator and denominator, and equation (4) simplifies to

$$\log \frac{\lambda_k(t|x_{[0,t)}^0)}{\lambda_k(t|x_{[0,t)}^0)+M\lambda_k^{\mathrm{q}}(t|x_{[0,t)}^0)} \text{ if } x_t^0 = k \text{ and } \log \frac{\lambda_k^{\mathrm{q}}(t|x_{[0,t)}^0)}{\lambda_k(t|x_{[0,t)}^0)+M\lambda_k^{\mathrm{q}}(t|x_{[0,t)}^0)} \text{ if } x_t^0 = \varnothing \qquad (5)$$

When a gradient-based optimization method adjusts $\theta$ to increase equation (5), the intuition is as follows. If $x_t^0 = k$, the model intensity $\lambda_k(t)$ is *increased* to explain why an event of type $k$ occurred at this particular time $t$. If $x_t^0 = \varnothing$, the model intensity $\lambda_k(t)$ is *decreased* to explain why an event of type $k$ did *not* actually occur at time $t$ (it was merely a noise event $x_t^m = k$, for some $m \neq 0$). These cases achieve the same qualitative effects as following the gradients of the first and second terms, respectively, in the log-likelihood (2).

Our full objective is an expectation of the sum of finitely many such log-ratios:[6]

$$J_{\mathrm{NC}}(\theta) \stackrel{\text{def}}{=} \mathbb{E}_{x_{[0,T)}^0 \sim p^*, x_{[0,T)}^{1:M} \sim q} \left[ \sum_{t:x_t^0 \neq \varnothing} \log \frac{\lambda_{x_t^0}(t|x_{[0,t)}^0)}{\underline{\lambda}_{x_t^0}(t|x_{[0,t)}^0)} + \sum_{m=1}^{M} \sum_{t:x_t^m \neq \varnothing} \log \frac{\lambda_{x_t^m}^{\mathrm{q}}(t|x_{[0,t)}^0)}{\underline{\lambda}_{x_t^m}(t|x_{[0,t)}^0)} \right] \qquad (6)$$

where $\underline{\lambda}_k(t \mid x_{[0,t)}^0) \stackrel{\text{def}}{=} \lambda_k(t \mid x_{[0,t)}^0) + M\lambda_k^{\mathrm{q}}(t \mid x_{[0,t)}^0)$. The expectation is estimated by sampling: we draw an observed stream $x_{[0,T)}^0$ from the training dataset, then draw noise events $x_{[0,T)}^{1:M}$ from $q$ conditioned on the prefixes (histories) given by this observed stream, as explained in the next section. Given these samples, the bracketed term is easy to compute (and we then use backprop to get its gradient w.r.t. $\theta$, which is a stochastic gradient of the objective (6)). It eliminates the $\int \sum$ of equation (2) as desired, replacing it with a sum over the noise events. For each real or noise event, we compute only two intensities—the true and noise intensities of that event type at that time.

### 3.1 Efficient Sampling of Noise Events

The **thinning algorithm** (Lewis & Shedler, 1979; Liniger, 2009) is a rejection sampling method for drawing an event stream over a given observation interval $[0, T)$ from a continuous-time autoregressive process. Suppose we have already drawn the first $i - 1$ times, namely $t_1, \ldots, t_{i-1}$. For every future time $t \geq t_{i-1}$, let $\mathcal{H}(t)$ denote the context $x_{[0,t)}$ consisting only of the events at those times, and define $\lambda(t \mid \mathcal{H}(t)) \stackrel{\text{def}}{=} \sum_{k=1}^{K} \lambda_k(t \mid \mathcal{H}(t))$. If $\lambda(t \mid \mathcal{H}(t))$ were constant at $\overline{\lambda}$, we could draw the next event time as $t_i \sim t_{i-1} + \mathrm{Exp}(\overline{\lambda})$. We would then set $x_t = \varnothing$ for all of the intermediate times $t \in (t_{i-1}, t_i)$, and finally draw the type $x_{t_i}$ of the event at time $t_i$, choosing $k$ with probability $\lambda_k(t_i \mid \mathcal{H}(t)) / \overline{\lambda}$. But what if $\lambda(t \mid \mathcal{H}(t))$ is not constant? The thinning algorithm still runs the foregoing method, taking $\overline{\lambda}$ to be any upper bound: $\overline{\lambda} \geq \lambda(t \mid \mathcal{H}(t))$ for all $t \geq t_{i-1}$. In this case, there may be "leftover" probability mass not allocated to any $k$. This mass is allocated to $\varnothing$. A draw of $x_{t_i} = \varnothing$ means there was no event at time $t_i$ after all (corresponding to a rejected proposal). Either way, we now continue on to draw $t_{i+1}$ and $x_{t_{i+1}}$, using a version of $\mathcal{H}(t)$ that has been updated to include the event or non-event $x_{t_i}$. The update to $\mathcal{H}(t)$ affects $\lambda(t \mid \mathcal{H}(t))$ and the choice of $\overline{\lambda}$.

**How to sample noise streams.** To draw a stream $x_{[0,t)}^m$ of noise events, we run the thinning algorithm, using the noise intensity functions $\lambda_k^{\mathrm{q}}$. However, there is a modification: $\mathcal{H}(t)$ is now defined to be $x_{[0,t)}^0$—the history from the *observed* event stream, rather than the previously sampled *noise* events—and is updated accordingly. This is because in equation (6), at each time $t$, all of $\{x_t^0, x_t^1, \ldots, x_t^M\}$ are conditioned on $x_{[0,t)}^0$ (akin to the discrete-time case).[7] The full pseudocode is given in Algorithm 1 in the supplementary material.

**Coarse-to-fine sampling of event types.** Although our NCE method has eliminated the need to integrate over $t$, the thinning algorithm above still sums over $k$ in the definition of $\lambda^{\mathrm{q}}(t \mid \mathcal{H}(t))$. For large $K$, this sum is expensive if we take the noise distribution on each training minibatch to

be, for example, the $p_\theta$ with the current value of $\theta$. That is a *statistically* efficient choice of noise distribution, but we can make a more *computationally* efficient choice. A simple scheme is to first generate each noise event with a coarse-grained type $c \in \{1, \ldots, C\}$, and then stochastically choose a refinement $k \in \{1, \ldots, K\}$:

$$\lambda_k^q(t \mid x_{[0,t)}^0) \stackrel{\text{def}}{=} \sum_{c=1}^{C} q(k \mid c)\lambda_c^q(t \mid x_{[0,t)}^0) \text{ for } k = 1, 2, \ldots, K \tag{7}$$

This noise model is parameterized by the functions $\lambda_c^q$ and the probabilities $q(k \mid c)$. The total intensity is now $\lambda^q(t \mid \mathcal{H}(t)) = \sum_{c=1}^{C} \lambda_c^q(t)$, so we now need to examine only $C$ intensity functions, not $K$, to choose $\overline{\lambda}$ in the thinning algorithm. If we *partition* the $K$ types into $C$ coarse-grained clusters (e.g., using domain knowledge), then evaluating the noise probability (7) within the training objective (6) is also fast because there is only one non-zero summand $c$ in equation (7). This simple scheme works well in our experiments. However, it could be elaborated by replacing $q(k \mid c)$ with $q(k \mid c, x_{[0,t)}^0)$, by partitioning the event vocabulary automatically, by allowing overlapping clusters, or by using multiple levels of refinement: all of these elaborations are used by the fast hierarchical language model of Mnih & Hinton (2009).

**How to draw $M$ streams.** An efficient way to draw the union of $M$ i.i.d. noise streams is to run the thinning algorithm once, with all intensities multiplied by $M$. In other words, the expected number of noise events on any interval is multiplied by $M$. This scheme does not tell us which specific noise stream $m$ generated a particular noise event, but the NCE objective (6) does not need to know that. The scheme works only because every noise stream $m$ has the same intensities $\lambda_k^q(t \mid x_{[0,t)}^0)$ (not $\lambda_k^q(t \mid x_{[0,t)}^m)$) at time $t$: there is no dependence on the previous events from that stream. Amusingly, NCE can now run even with non-integer $M$.

**Fractional objective.** One view of the thinning algorithm is that it accepts the proposed time $t_i$ with probability $\mu = \lambda(t_i)/\overline{\lambda}$, and in that case, labels it as $k$ with probability $\lambda_k(t_i)/\lambda(t_i)$. To get a greater diversity of noise samples, we can accept the time with probability 1, if we then scale its term in the objective (6) by $\mu$. This does not change the expectation (6) but may reduce the sampling variance in estimating it. Note that increasing the upper bound $\overline{\lambda}$ now has an effect similar to increasing $M$: more noise samples.[8]

## 3.2 Computational Cost Analysis

State-of-the-art intensity models use neural networks whose state summarizes the history and is updated after each event. So to train on a single event stream $x$ with $I \geq 0$ events, both MLE and NCE must perform $I$ updates to the neural state. Both MLE and NCE then evaluate the intensities $\lambda_k(t \mid x_{[0,t)})$ of these $I$ events, and also the intensities of a number of events that did *not* occur, which almost surely fall at other times.[9]

Consider the *number of intensities evaluated*. For MLE, assume the Monte Carlo integration technique mentioned in section 2.2. MLE computes the intensity $\lambda$ for $I$ observed events and for all $K$ possible events at each of $J$ sampled times. We take $J = \rho I$ (with randomized rounding to an integer), where $\rho > 0$ is a hyperparameter (Mei & Eisner, 2017). Hence, the expected total number of intensity evaluations is $I + \rho I K$.

For NCE with the coarse-to-fine strategy, let $J$ be the total number of times *proposed* by the thinning algorithm. Observe that $\mathbb{E}[I] = \int_0^T \lambda^*(t \mid x_{[0,t)})dt$, and $\mathbb{E}[J] = M \cdot \int_0^T \overline{\lambda}(t \mid x_{[0,t)})dt$. Thus, $\mathbb{E}[J] \approx M \cdot \mathbb{E}[I]$ if (1) $\overline{\lambda}$ at any time is a tight upper bound on the noise event rate $\lambda^q$ at that time and (2) the average noise event rate well-approximates the average observed event rate (which should become true very early in training). To label or reject each of the $J$ proposals, NCE evaluates $C$ noise intensities $\lambda_c^q$; if the proposal is accepted with label $k$ (perhaps fractionally), it must also evaluate its model intensity $\lambda_k$. The noise and model intensities $\lambda_c^q$ and $\lambda_k$ must also be evaluated for the $I$ observed events. Hence, the total number of intensity evaluations is at most $(C+1)J + 2I$, which $\approx (C+1)MI + 2I$ in expectation.

Dividing by $I$, we see that making $(M+1)(C+1) \leq \rho K$ suffices to make NCE's stochastic objective take less work per observed stream than MLE's stochastic objective. $M = 1$ and $C = 1$ is a valid choice. But NCE's objective is less informed for smaller $M$, so its stochastic gradient

carries less information about $\theta^*$. In section 5, we empirically investigate the effect of $M$ and $C$ on NCE and compare to MLE with different $\rho$.

### 3.3 Theoretical Guarantees: Optimality, Consistency and Efficiency

The following theorem implies that stochastic gradient ascent on NCE converges to a correct $\theta$ (if one exists):

**Theorem 1** (Optimality). *Under assumptions 1 and 2, $\theta \in \mathrm{argmax}_\theta J_{\mathrm{NC}}(\theta)$ if and only if $p_\theta = p^*$.*

This theorem falls out naturally when we rearrange the NCE objective in equation (6) as

$$\int_{t=0}^{T} \sum_{x^0_{[0,t)}} p^*(x^0_{[0,t)}) \sum_{k=1}^{K} \underline{\lambda}^*_k(t \mid x^0_{[0,t)}) \underbrace{\left( \frac{\lambda^*_k(t|x^0_{[0,t)})}{\underline{\lambda}^*_k(t|x^0_{[0,t)})} \log \frac{\lambda_k(t|x^0_{[0,t)})}{\underline{\lambda}_k(t|x^0_{[0,t)})} + M \frac{\lambda^q_k(t|x^0_{[0,t)})}{\underline{\lambda}^*_k(t|x^0_{[0,t)})} \log \frac{\lambda^q_k(t|x^0_{[0,t)})}{\underline{\lambda}_k(t|x^0_{[0,t)})} \right)}_{\text{a negative cross entropy}} dt$$

where $\lambda^*_k$ is the intensity under $p^*$ and $\underline{\lambda}^*_k$ is defined analogously to $\underline{\lambda}_k$: see full derivation in Appendix B.1. Obviously, $p_\theta = p^*$ is *sufficient* to maximize the negative cross-entropy for any $k$ given any history and thus maximize $J_{\mathrm{NC}}(\theta)$. It turns out to be also *necessary* because any $\theta$ for which $p_\theta \neq p^*$ would, given assumption 1, end up decreasing the negative cross-entropy for some $k$ over some interval $(t, t')$ given a set of histories with non-zero measure. A full proof can be found in Appendix B.2: as we'll see there, although it resembles Theorem 3.2 of Ma & Collins (2018), the proof of our Theorem 1 requires new analysis to handle continuous time, since Ma & Collins (2018) only worked on discrete-time sequential data.

Moreover, our NCE method is strongly consistent for any $M \geq 1$ and approaches *Fisher efficiency* when $M$ is large. These properties are the same as in Ma & Collins (2018) and the proofs are also similar. Therefore, we leave the related theorems together with their assumptions and proofs to Appendices B.3 and B.4.

## 4 Related Work

The original "binary classification" NCE principle was proposed by Gutmann & Hyvärinen (2010) to estimate parameters for joint models of the form $p_\theta(x) \propto \exp(\mathrm{score}(x, \theta))$. Gutmann & Hyvärinen (2012) applied it to natural image statistics. It was then widely applied to natural language processing problems such as language modeling (Mnih & Teh, 2012), learning word representations (Mikolov et al., 2013) and machine translation (Vaswani et al., 2013). The "ranking-based" variant (Jozefowicz et al., 2016)[10] is better suited for conditional distributions (Ma & Collins, 2018), including those used in autoregressive models, and has shown strong performance in large-scale language modeling with recurrent neural networks.

Guo et al. (2018) tried NCE on (univariate) point processes but used the binary classification version. They used discrimination problems of the form: "Is event $k$ at time $t'$ the true next event following history $x_{[0,t]}$, or was it generated from a noise distribution?" Their classification-based NCE variant is *not* well-suited to conditional distributions (Ma & Collins, 2018): this complicates their method since they needed to build a parametric model of the local normalizing constant, giving them weaker theoretical guarantees and worse performance (see section 5). In contrast, we choose the ranking-based variant: our key idea of how to apply this to continuous time is new (see section 3) and requires new analysis (see Appendices A and B).

## 5 Experiments

We evaluate our NCE method on several synthetic and real-world datasets, with comparison to MLE, Guo et al. (2018) (denoted as b-NCE), and least-squares estimation (LSE) (Eichler et al., 2017). b-NCE has the same hyper-parameter $M$ as our NCE, namely the number of noise events. LSE's objective involves an integral over times $[0, T)$, so it has the same hyper-parameter $\rho$ as MLE.

On each of the datasets, we will show the estimated log-likelihood on the held-out data achieved by the models trained on the NCE, b-NCE, MLE and LSE objectives, as training consumes increasing amounts of computation—measured by the number of intensity evaluations and the elapsed wall-clock time (in seconds).[11] We always set the minibatch size $B$ to exhaust the GPU capacity, so smaller $\rho$ or $M$ allows larger $B$. Larger $B$ in turn increases the number of epochs per unit time (but decreases the possibly beneficial variance in the stochastic gradient updates).

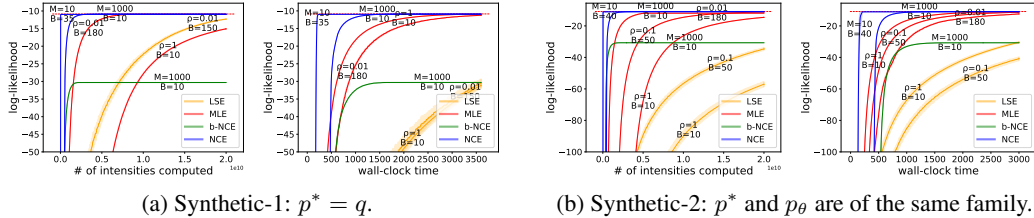

(a) Synthetic-1: $p^* = q$.

(b) Synthetic-2: $p^*$ and $p_\theta$ are of the same family.

Figure 1: Learning curves of MLE and NCE on synthetic datasets. The displayed $\rho$ and $M$ values are among the better ones that we found during hyperparameter search. The horizontal red line marks the highest held-out log-likelihood achieved by MLE. The shaded area of each curve shows the range of log-likelihood of three independent runs; most of them are too narrow to be easily noticed.

## 5.1 Synthetic Datasets

In this section, we work on two synthetic datasets with $K = 10000$ event types. We choose the **neural Hawkes process (NHP)** (Mei & Eisner, 2017) to be our model $p_\theta$.[12] For the noise distribution $q$, we choose $C = 1$ and also parametrize its intensity function as a neural Hawkes process.

The first dataset has sequences drawn from the randomly initialized $q$ such that we can check how well our NCE method could perform with the "ground-truth" noise distribution $q = p^*$; the sequences of the second dataset were drawn from a randomly initialized neural Hawkes process to evaluate both methods in the case that the model family $p_\theta$ is well-specified. We show (the zoomed-in views of the interesting parts of) multiple learning curves on each dataset in Figure 1: NCE is observed to consume substantially fewer intensity evaluations and less wall-clock time than MLE to achieve competitive log-likelihood, while b-NCE and LSE are slower and only converge to lower log-likelihood. Note that the wall-clock time may not be proportional to the number of intensities because computing intensities is not all of the work (e.g., there are LSTM states of both $p_\theta$ and $q$ to compute and store on GPU).

We also observed that models that achieved comparable log-likelihood—no matter how they were trained—achieved comparable prediction accuracies (measured by root-mean-square-error for time and error rate for type). Therefore, our NCE still beats other methods at converging quickly to the highest prediction accuracy.

**Ablation Study I: Always or Never Redraw Noise Samples.** During training, for each observed data, we can choose to either redraw a new set of noise samples every time we train on it or keep reusing the old samples: we did the latter for Figure 1. In experiments doing the former, we observed better generation for tiny $M$ (e.g., $M = 1$) but substantial slow-down (because of sampling) with no improved generalization for large $M$ (e.g, 1000). Such results suggest that we always reuse old samples as long as $M$ is reasonably large: it is then what we do for all other experiments throughout the paper. See Appendix D.4 for more details of this ablation study, including learning curves of the "always redraw" strategy in Figure 5.

## 5.2 Real-World Social Interaction Datasets with Large $K$

We also evaluate the methods on several real-world social interaction datasets that have many event types: see Appendix D.1 for details (e.g, data statistics, pre-processing, data splits, etc). In this section, we show the learning curves on two particularly interesting datasets (explained below) in Figure 2 and leave those on the other datasets (which look similar) to Appendix D.3.

**EuroEmail** (Paranjape et al., 2017). This dataset contains time-stamped emails between anonymized members of a European research institute. We work on a subset of 100 most active members and then end up with $K = 10000$ possible event types and 50000 training event tokens.

**BitcoinOTC** (Kumar et al., 2016). This dataset contains time-stamped rating (positive/negative) records between anonymized users on the BitcoinOTC trading platform. We work on a subset of 100 most active users and then end up with $K = 19800$ (self-rating not allowed) possible event types but only 1000 training event tokens: this is an extremely data-sparse setting.

On these datasets, our model $p_\theta$ is still a neural Hawkes process. For the noise distribution $q$, we experiment with not only the coarse-to-fine neural process with $C = 1$ but also a homogeneous Poisson

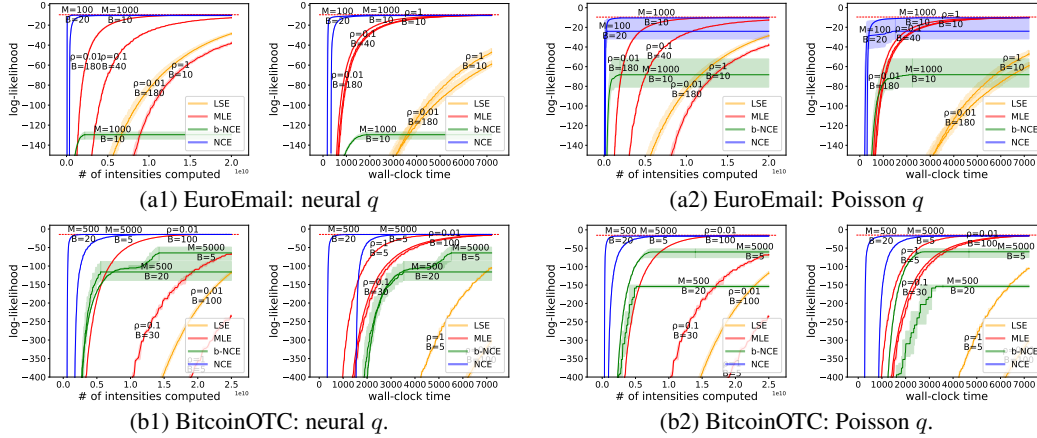

Figure 2: Learning curves of MLE and NCE on the real-world social interaction datasets.

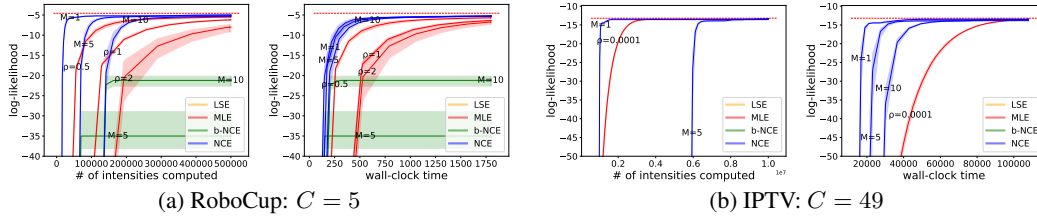

Figure 3: Learning curves of MLE and NCE on RoboCup and IPTV datasets.

process. As shown in Figure 2, our NCE tends to perform better with the neural $q$: this is because a neural model can better fit the data and thus provide better training signals, analogous to how a good generator can benefit the discriminator in the generative adversarial framework (Goodfellow et al., 2014). NCE with Poisson $q$ also shows benefits through the early and middle training stages, but it might suffer larger variance (e.g., Figure 2a2) and end up with slightly worse generalization (e.g., Figure 2b2). MLE with different $\rho$ values all eventually achieve the highest log-likelihood ($\approx -10$ on EuroEmail and $\approx -15$ on BitcoinOTC), but most of these runs are so slow that their peaks are out of the current views. The b-NCE runs with different $M$ values are slower, achieve worse generalization and suffer larger variance than our NCE; interestingly, b-NCE prefers Poisson $q$ to neural $q$ (better generalization on EuroEmail and smaller variance on BitcoinOTC). In general, LSE is the slowest, and the highest log-likelihood it can achieve ($\approx -30$ on EuroEmail and $\approx -25$ on BitcoinOTC) is lower than that of MLE and our NCE.

**Ablation Study II: Trained vs. Untrained $q$.** The noise distributions (except the ground-truth $q$ for Synthetic-1) that we have used so far were all pretrained on the same data as we train $p_\theta$. The training cost is cheap: e.g., on the datasets in this section, the actual wall-clock training time for the neural $q$ is less than 2% of what is needed to train $p_\theta$, and training the Poisson $q$ costs even less.[13][14] We also experimented with untrained noise distributions and they were observed to perform worse (e.g., worse generalization, slower convergence and larger variance). See Appendix D.5 for more details, including learning curves (Figure 6).

## 5.3 Real-World Dataset with Dynamic Facts

In this section, we let $p_\theta$ be a **neural Datalog through time (NDTT)** model (Mei et al., 2020). Such a model can be used in a domain in which new events dynamically update the set of event types and the structure of their intensity functions. We evaluate our method on training the domain-specific models presented by Mei et al. (2020), on the same datasets they used:

**RoboCup** (Chen & Mooney, 2008). This dataset logs actions of robot players during RoboCup soccer games. The set of possible event types dynamically changes over time (e.g., only ball possessor can kick or pass) as the ball is frequently transferred between players (by passing or stealing). There are $K = 528$ event types over all time, but only about 20 of them are possible at any given time.

**IPTV** (Xu et al., 2018). This dataset contains time-stamped records of 1000 users watching 49 TV programs over 2012. The users are not able to watch a program until it is released, so the number of event types grows from $K = 0$ to $K = 49000$ as programs are released one after another.

The learning curves are displayed in Figure 3. On RoboCup, NCE only progresses faster than MLE at the early to middle training stages: $M = 5$ and $M = 10$ eventually achieved the highest log-likelihood at the same time as MLE and $M = 1$ ended up with worse generalization. On IPTV, NCE with $M = 1$ turned out to learn as well as and much faster than MLE. The dynamic architecture makes it hard to parallelize the intensity computation; MLE in particular performs poorly in wall-clock time, and we needed a remarkably small $\rho$ to let MLE finish within the shown time range. On both datasets, b-NCE and LSE drastically underperform MLE and NCE: their learning curves increase so slowly and achieve such poor generalization that only b-NCE with $M = 5$ and $M = 10$ are visible on the graphs.

**Ablation Study III: Effect of** $C$**.** In the above figures, we used the coarse-to-fine neural model as $q$. On RoboCup, each action (kick, pass, etc.) has a coarse-grained intensity, so $C = 5$. On IPTV, we partition the event vocabulary by TV program, so $C = 49$. We also experimented with $C = 1$: this reduces the number of intensities computed during sampling on both datasets, but has (slightly) worse generalization on RoboCup (since $q$ becomes less expressive). See Appendix D.6 for more details, including learning curves (Figure 7).

## 6    Conclusion

We have introduced a novel instantiation of the general NCE principle for training a multivariate point process model. Our objective has the same optimal parameters as the log-likelihood objective (if the model is well-specified), but needs fewer expensive function evaluations and much less wall-clock time in practice. This benefit is demonstrated on several synthetic and real-world datasets. Moreover, our method is provably consistent and efficient under mild assumptions.

## Broader Impact

Our method is designed to train a multivariate point process for probabilistic modeling of event streams. By describing this method and releasing code, we hope to facilitate probabilistic modeling of continuous-time sequential data in many domains. Good probabilistic models make it possible to impute missing events, anticipate possible future events, and react accordingly. They can also be used in exploratory data analysis.

In addition to making it more feasible and more convenient for domain experts to train complex models with many event types, our method reduces the energy cost necessary to do so.

Examples of event streams with potential social impact include a person's detailed food/exercise/sleep/medical event log, their social media interactions, their interactions with educational exercises or games, or their educational or workplace events (for time management and career planning); a customer's interactions with a particular company or its website or other user interface; a company's sales and purchases; geopolitical events, financial events, human activity modeling, music modeling, and dynamic resource requests.

We are not aware of any negative broader impacts that might stem from publishing this work.

## Disclosure of Funding Sources

This work was supported by a Ph.D. Fellowship Award to the first author by Bloomberg L.P. and a National Science Foundation Grant No. 1718846 to the last author, as well as two Titan X Pascal GPUs donated by NVIDIA Corporation and compute cycles from the Maryland Advanced Research Computing Center.

## Acknowledgments

We thank the anonymous NeurIPS reviewers and meta-reviewer as well as Hongteng Xu for helpful comments on this paper.

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
