[Supplementary Material]

# Notes

[1]A special event $x_0$ is sometimes given at time 0 to mark the beginning of the sequence; the model then generates the rest of the sequence conditioned on $x_0$.

[2]The product $p^*(x_t^m \mid x_{0:t}) \prod_{m' \neq m} q(x_t^{m'} \mid x_{0:t})$ is the likelihood of $x_t^m$ being the one drawn from $p^*$. The prior is uniform since any $m$ in the unordered bag was *a priori* equally probable.

[3]In practice, it is more convenient to maximize the expected *sum* over $t$ in a sequence drawn uniformly from the set of sequences in the training dataset. This scales the objective up by the average sequence length, preserving the property that longer sequences have more weight.

[4]Our model does not need any normalization: $p(x_t = \varnothing) + \sum_{k=1}^{K} p(x_t = k) = 1 +$ (infinitesimal quantities) $= 1$.

[5] While this paper's speedup over the MLE objective (2) comes from avoiding the integral, an alternative would be to estimate the integral more efficiently. One might try randomized adaptive quadrature (Baran et al., 2008) modified for our discontinuous intensity functions and GPU hardware; or importance sampling of $(t, k)$ pairs where the proposal distribution is roughly proportional to $\lambda_k(t)$—much like the noise distribution we will develop for NCE.

[6]We remark that $J_{\mathrm{NC}}(\theta)$ is the expected log-*probability* of a discrete choice, whereas $J_{\mathrm{LL}}(\theta)$ was the expected log-*density* of an observation that includes continuous times. A density must be integrated to yield a probability.

[7]This is not essential to the NCE approach, since in principle the $M + 1$ elements of the bag could all be drawn from different distributions. However, the homogeneity simplifies equations (5)–(6), and not having to keep track of previous noise samples simplifies bookkeeping. Furthermore, much as in a GAN, we expect the discrimination task to be most challenging and informative when the noise intensity $\lambda_k^{\mathrm{q}}$ at time $t$ is close to the true intensity $\lambda_k^*(t \mid x_{[0,t)}^0)$. Therefore we give the function $\lambda_k^{\mathrm{q}}$ access to the true history $x_{[0,t)}^0$, and will train it to predict something like the true intensity.

[8]This trick does carry computational cost: we need to train (via backpropagation) on proposals that might not have been accepted otherwise. This cost is perhaps not worth it when $\mu(t)$ is too low: it might be better spent on increasing $M$ or running more training epochs for a fixed $M$. As a compromise, if $\mu$ is small ($\leq 0.05$ in our current experiments), we revert to the original approach of accepting the time with probability $\mu$ and not scaling it.

[9]In between the events, even if the neural state remains constant, the intensity functions need not be constant.

[10]Jozefowicz et al. (2016) considered it a competitor to NCE; Ma & Collins (2018) argued for regarding it as a variant.

[11]Our code is written in PyTorch (Paszke et al., 2017) and will be released upon paper acceptance. Our experiments were run on NVIDIA Tesla K80.

[12]We use the public PyTorch implementation. NHP is a thoughtfully designed framework that has been demonstrated effective on temporal data, but our method can also be used for other models with parametric intensity functions.

[13]We train $q$ by MLE: summing $C$ intensities is not expensive when $C$ is small. In Appendix C.2, we document an alternative strategy that uses $q$ as the noise distribution to train itself by NCE.

[14]For the experiments in section 5.3, training the neural $q$ takes only $< 1/100$ of what needed to train $p_\theta$.

# Appendices

## A  Proof Details for MLE

In this section, we prove the claim in section 2.2 that $\arg\max_\theta J_{\text{LL}}(\theta) = \Theta^* \stackrel{\text{def}}{=} \{\theta^* : p_{\theta^*} = p^*\}$. For this purpose, we first rearrange $J_{\text{LL}}(\theta) = \mathbb{E}_{p^*(x_{[0,T)})}\left[\log p_\theta(x_{[0,T)})\right]$ as below:

$$\sum_{x_{[0,T)}} p^*(x_{[0,T)}) \log p_\theta(x_{[0,T)}) \tag{8a}$$

$$= \int_{t=0}^{T} \sum_{x_{[0,t)}} p^*(x_{[0,t)}) \underbrace{\sum_{x_{[t,t+dt)}} p^*(x_{[t,t+dt)} \mid x_{[0,t)}) \log p_\theta(x_{[t,t+dt)} \mid x_{[0,t)})}_{\text{call it } H_\theta(t,x_{[0,t)})} \tag{8b}$$

The intuition for equation (8b) is that due to the form of the autoregressive model, $\log p_\theta(x_{[0,T)})$ in equation (8a) can be broken up into a sum of log (infinitesimal) probabilities of $x_{[t,t+dt)}$ on the infinitesimal intervals $[t, t+dt)$, each probability being conditioned on the past history $x_{[0,t)}$. When we take the expectation under $p^*$, each summand gets weighted by the probability that $x_{[0,t)}$ and $x_{[t,t+dt)}$ would take on the values in that summand. This gives a form (8b) that aggregates the infinitesimal quantities $H_\theta(t, x_{[0,t)})$ over possible times $t \in [0, T)$ and possible histories $x_{[0,t)}$.

*Proof.* We first observe that $H_\theta(t, x_{[0,t)})$ is the negative cross-entropy between the conditional distributions of $p^*$ and $p_\theta$ at time $t$ (both conditioned on history $x_{[0,t)}$). Technically, $x_{[t,t+dt)}$ will have an event of type $k$ with probability $\lambda_k^*(t)dt$ under $p^*$ ($\lambda_k(t)dt$ under $p_\theta$) or has no event at all with probability $1 - \sum_{k=1}^{K} \lambda_k^*(t)dt$ under $p^*$ ($1 - \sum_{k=1}^{K} \lambda_k(t)dt$ under $p_\theta$). So the term $H_\theta(t, x_{[0,t)})$ is actually the negative cross entropy between the following two discrete distributions over $\{\varnothing, 1, \ldots, K\}$:

$$\left[\left(1 - \sum_{k=1}^{K} \lambda_k^*(t \mid x_{[0,t)})dt\right), \ \lambda_1^*(t \mid x_{[0,t)})dt, \ \ldots, \ \lambda_K^*(t \mid x_{[0,t)})dt\right] \tag{9a}$$

$$\left[\left(1 - \sum_{k=1}^{K} \lambda_k(t \mid x_{[0,t)})dt\right), \ \lambda_1(t \mid x_{[0,t)})dt, \ \ldots, \ \lambda_K(t \mid x_{[0,t)})dt\right] \tag{9b}$$

The (infinitesimal) negative cross-entropy between them is always smaller than or equal to the negative entropy of the distribution in equation (9a): it will be strictly smaller if these two distributions are distinct, and equal when they are identical.

It is then obvious that any $\theta^* \in \Theta^*$ maximizes $J_{\text{LL}}(\theta)$ because it maximizes the negative cross-entropy for any history $x_{[0,t)}$ at any time $t$.

To check if any other $\bar{\theta} \notin \Theta^*$ maximizes $J_{\text{LL}}(\theta)$ as well, we analyze

$$J_{\text{LL}}(\bar{\theta}) - J_{\text{LL}}(\theta^*) = \int_{t=0}^{T} \sum_{x_{[0,t)}} p^*(x_{[0,t)}) \underbrace{(H_{\bar{\theta}}(t, x_{[0,t)}) - H_{\theta^*}(t, x_{[0,t)}))}_{\text{denote it as } G_{\bar{\theta}}(t, x_{[0,t)})dt} \tag{10}$$

where $\theta^*$ can be any member in $\Theta^*$. Note that we can denote $H_{\bar{\theta}} - H_{\theta^*}$ as $G_{\bar{\theta}}dt$ because the probabilities in $H$ and thus the entropy changes (if any) are all infinitesimal.

According to the definition of $\bar{\theta}$ and $\theta^*$, there must exist a stream $\bar{x}_{[0,T)}$, a time $\bar{t} \in (0, T)$ and a type $\bar{k} \in \{1, \ldots, K\}$ such that $\lambda_{\bar{k}}(\bar{t} \mid \bar{x}_{[0,\bar{t})}) \neq \lambda_{\bar{k}}^*(\bar{t} \mid \bar{x}_{[0,\bar{t})})$. Therefore, we have $G_{\bar{\theta}}(\bar{t}, \bar{x}_{[0,\bar{t})}) < 0$ since the distributions in equation (9) are distinct for the given history $\bar{x}_{[0,\bar{t})}$. Does this difference lead to any overall change of the entire objective?

Actually, according to Lemma 1 (that we will prove shortly), the existence of such $\bar{x}_{[0,T)}$, $\bar{t}$ and $\bar{k}$ implies that there exists an interval $(t', t'') \subset [0, T)$ such that, for any $t \in (t', t'')$, there exists a set $\mathcal{X}(t)$ of histories with non-zero measure such that any $x_{[0,t)} \in \mathcal{X}(t)$ satisfies $\lambda_{\bar{k}}(t \mid x_{[0,t)}) \neq \lambda_{\bar{k}}^*(t \mid$

$x_{[0,t)}$). That is to say, the fraction of the integral over $(t', t'')$ is a non-infinitesimal negative number:

$$\int_{t=t'}^{t''} \sum_{x_{[0,t)}} p^*(x_{[0,t)}) G_{\bar{\theta}}(t, x_{[0,t)}) dt \tag{11a}$$

$$= \underbrace{\int_{t=t'}^{t''} \sum_{x_{[0,t)} \in \mathcal{X}(t)} p^*(x_{[0,t)}) G_{\bar{\theta}}(t, x_{[0,t)}) dt}_{<0} + \underbrace{\int_{t=t'}^{t''} \sum_{x_{[0,t)} \notin \mathcal{X}(t)} p^*(x_{[0,t)}) G_{\bar{\theta}}(t, x_{[0,t)}) dt}_{\leq 0} \tag{11b}$$

where the second integral $\leq 0$ because $G_\theta$ always $\leq 0$. For the same reason, we also have $\int_{t=0}^{t'} \sum_{x_{[0,t)}} p^*(x_{[0,t)}) G_{\bar{\theta}}(t, x_{[0,t)}) dt \leq 0$ and $\int_{t=t''}^{T} \sum_{x_{[0,t)}} p^*(x_{[0,t)}) G_{\bar{\theta}}(t, x_{[0,t)}) dt \leq 0$. Then the overall difference must be strictly negative, i.e.,

$$J_{\text{LL}}(\bar{\theta}) - J_{\text{LL}}(\theta^*) < 0 \tag{12}$$

Note that this inequality holds for any $\bar{\theta} \notin \Theta^*$ and any $\theta^* \in \Theta^*$, meaning that $\theta^* \in \Theta^*$ is necessary to maximize the objective.

Now the proof of $\text{argmax}_\theta J_{\text{LL}}(\theta) = \Theta^*$ is complete.

$\square$

**Lemma 1.** *Suppose that we have two intensity functions that meet assumption 1: they have different parameters $\theta$ and $\theta^*$ and are denoted as $\lambda_k(t \mid x_{[0,t)})$ and $\lambda_k^*(t \mid x_{[0,t)})$ respectively. If there exists a stream $\bar{x}_{[0,T)}$, a time $\bar{t} \in (0, T)$ and a type $\bar{k} \in \{1, \ldots, K\}$ such that $\lambda_{\bar{k}}(\bar{t} \mid \bar{x}_{[0,\bar{t})}) \neq \lambda_{\bar{k}}^*(\bar{t} \mid \bar{x}_{[0,\bar{t})})$, then there exists an open interval $(t', t'') \subset [0, T)$ such that, for any $t \in (t', t'')$, there exists a set $\mathcal{X}$ of histories with non-zero measure such that any $x_{[0,t)} \in \mathcal{X}$ satisfies $\lambda_{\bar{k}}(t \mid x_{[0,t)}) \neq \lambda_{\bar{k}}^*(t \mid x_{[0,t)})$.*

This lemma says: if $\theta$ and $\theta^*$ are meaningfully different in that they predict different intensities at time $t$ for *some history*, then they actually do so for a *set of histories of non-zero measure*, making this difference visible in the objective functions like $J_{\text{LL}}(\theta)$ (see above) and $J_{\text{NC}}(\theta)$ (see Appendix B). Note that previous work did not encounter this since they only worked on either non-sequential data (e.g., Gutmann & Hyvärinen (2010, 2012)) or discrete-time sequential data (e.g., Ma & Collins (2018)).

*Proof.* We first prove the existence of an interval $(t', t'')$ such that $\lambda_{\bar{k}}(t \mid \bar{x}_{[0,t)}) \neq \lambda_{\bar{k}}^*(t \mid \bar{x}_{[0,t)})$ for the given stream $\bar{x}_{[0,T)}$ and any time $t \in (t', t'')$. It turns out to be straightforward under assumption 1: since the intensity functions are continuous between events, we can construct this interval by expanding from the given time $\bar{t}$ until $\lambda_{\bar{k}}(t \mid \bar{x}_{[0,t)}) = \lambda_{\bar{k}}^*(t \mid \bar{x}_{[0,t)})$.

We use $d$ to denote the maximal difference between the intensities over $(t', t'')$, i.e., $d \stackrel{\text{def}}{=} \max_{t \in (t', t'')} |\lambda_{\bar{k}}(t \mid \bar{x}_{[0,t)}) - \lambda_{\bar{k}}^*(t \mid \bar{x}_{[0,t)})|$. Then, to facilitate the rest of the proof, we shrink the interval $(t', t'')$ such that $|\lambda_{\bar{k}}(t \mid \bar{x}_{[0,t)}) - \lambda_{\bar{k}}^*(t \mid \bar{x}_{[0,t)})| > d/2$ for any time $t \in (t', t'')$.

Now, for any time $t \in (t', t'')$, we prove the existence of the set described in Lemma 1 by constructing it.

We initialize this set as $\{\bar{x}_{[0,t)}\}$. If $\bar{x}_{[0,t)}$ doesn't have any event, then its probability $p(\bar{x}_{[0,t)}) = \exp(-\int_{s=0}^{t} \sum_{k=1}^{K} \lambda_k(s \mid \bar{x}_{[0,s)}) ds)$ is not infinitesimal and this set already has non-zero measure.

What if $\bar{x}_{[0,t)}$ has $I \geq 1$ events at times $0 < t_1 < \ldots < t_I < t$? Intuitively, we can construct many other histories satisfying the intensity inequality by slightly shifting the time of each event: as long as they aren't shifted by too far, the $d/2$ difference between intensities won't vanish (even if it decreases). See the formal proof as below.

In the case of $I \geq 1$, the probability $p(\bar{x}_{[0,t)})$ is infinitesimal in the order of $(dt)^I$: $p(\bar{x}_{[0,t)}) = \prod_{i=1}^{I} (\lambda_{\bar{x}_{t_i}}(t_i \mid \bar{x}_{[0,t_i)}) dt) \exp(-\int_{s=0}^{t} \sum_{k=1}^{K} \lambda_k(s \mid \bar{x}_{[0,s)}) ds)$. Therefore, to construct a set with non-zero measure, the number of histories satisfying the inequality has to be in the order of $(\frac{1}{dt})^I$.

We define an open interval $(t_1', t_1'')$ that covers $t_1$ but not any other event time. Now we can construct uncountably many—in the order of $\frac{1}{dt}$—histories $x_{[0,t)}$ by freely shifting the event time $t_1$ inside $(t_1', t_1'')$. Suppose that $t_1$ has been shifted by $\Delta \in \mathbb{R}$. Under assumption 1, there is a continuous function $c(\Delta)$ such that $c(0) = 0$ and

$$\lambda_{\bar{k}}(t \mid x_{[0,t)}) - \lambda_k^*(t \mid x_{[0,t)}) = \lambda_{\bar{k}}(t \mid \bar{x}_{[0,t)}) - \lambda_k^*(t \mid \bar{x}_{[0,t)}) + c(\Delta) \tag{13}$$

meaning that the intensity difference will change by $c(\Delta)$. By triangle inequality, we have

$$\left| \lambda_{\bar{k}}(t \mid x_{[0,t)}) - \lambda_k^*(t \mid x_{[0,t)}) \right| \geq \left| \left| \lambda_{\bar{k}}(t \mid \bar{x}_{[0,t)}) - \lambda_k^*(t \mid \bar{x}_{[0,t)}) \right| - |c(\Delta)| \right| \tag{14}$$

Since $c(\Delta)$ is continuous, as long as we make $|\Delta|$ small enough, we'll have $|c(\Delta)| \leq d/2$ and then the following inequality holds:

$$\left| \lambda_{\bar{k}}(t \mid x_{[0,t)}) - \lambda_k^*(t \mid x_{[0,t)}) \right| \geq \left| \lambda_{\bar{k}}(t \mid \bar{x}_{[0,t)}) - \lambda_k^*(t \mid \bar{x}_{[0,t)}) \right| - |c(\Delta)| > d/2 - d/2 = 0 \tag{15}$$

meaning that the intensities given the new history are still different. Therefore, as long as we keep the interval $(t_1', t_1'')$ small enough, we'll have order-$\frac{1}{dt}$ many histories and the inequality in equation (15) holds given any of them.

Recall that we need order-$(\frac{1}{dt})^I$ many such histories. We can obtain them by simply defining $I$ *disjoint* open intervals $(t_1', t_1''), \ldots, (t_I', t_I'')$ such that $t_i \in (t_i', t_i'')$ and freely shifting each event time $t_i$ inside $(t_i', t_i'')$. Suppose that $t_i$ has been shifted by $\Delta_i \in \mathbb{R}$, Under assumption 1, there is a continuous function $c(\Delta_1, \ldots, \Delta_I)$ such that $c(0, \ldots, 0) = 0$ and

$$\lambda_{\bar{k}}(t \mid x_{[0,t)}) - \lambda_k^*(t \mid x_{[0,t)}) = \lambda_{\bar{k}}(t \mid \bar{x}_{[0,t)}) - \lambda_k^*(t \mid \bar{x}_{[0,t)}) + c(\Delta_1, \ldots, \Delta_I) \tag{16}$$

Since $c$ is a continuous function, there exist $I$ positive real numbers $\bar{\Delta}_1, \ldots, \bar{\Delta}_I$ such that $|c(\Delta_1, \ldots, \Delta_I)| \leq d/2$ as long as $|\Delta_i| \leq \bar{\Delta}_i$ holds for all $i = 1, \ldots, I$. In this case, by triangle inequality, we still have

$$\left| \lambda_{\bar{k}}(t \mid x_{[0,t)}) - \lambda_k^*(t \mid x_{[0,t)}) \right| \geq \left| \lambda_{\bar{k}}(t \mid \bar{x}_{[0,t)}) - \lambda_k^*(t \mid \bar{x}_{[0,t)}) \right| - |\Delta_i| > 0 \tag{17}$$

Now we have order-$(\frac{1}{dt})^I$ many histories: each of them has order-$(dt)^I$ probability and the inequality in equation (17) holds given any of them. That is to say, the set of these histories has non-zero measure and we have $\lambda_{\bar{k}}(t \mid x_{[0,t)}) \neq \lambda_k^*(t \mid x_{[0,t)})$ given any $x_{[0,t)}$ in this set.

This completes the proof.

$\square$

# B    NCE Details

In this section, we will discuss the theoretical guarantees of our NCE method in detail.

## B.1    Derivation Details

In this section, we show how to get the rearranged NCE objective in section 3.3 from equation (6).

First of all, we observe that:

$$\mathbb{E}_{x_{[0,T)}^0 \sim p^*, x_{[0,T)}^{1:M} \sim q} \left[ \sum_{t:x_t^0 \neq \varnothing} \log \frac{\lambda_{x_t^0}(t \mid x_{[0,t)}^0)}{\lambda_{x_t^0}^q(t \mid x_{[0,t)}^0)} + \sum_{m=1}^M \sum_{t:x_t^m \neq \varnothing} \log \frac{\lambda_{x_t^m}^q(t \mid x_{[0,t)}^0)}{\lambda_{x_t^m}(t \mid x_{[0,t)}^0)} \right] \tag{18a}$$

$$= \int_{t=0}^T \mathbb{E}_{x_{[0,t)}^0 \sim p^*} \left[ \sum_{k=1}^K \lambda_k^*(t \mid x_{[0,t)}^0) dt \log \frac{\lambda_{x_t^0}(t \mid x_{[0,t)}^0)}{\lambda_{x_t^0}^q(t \mid x_{[0,t)}^0)} + \sum_{m=1}^M \sum_{k=1}^K \lambda_k^q(t \mid x_{[0,t)}^0) dt \log \frac{\lambda_{x_t^m}^q(t \mid x_{[0,t)}^0)}{\lambda_{x_t^m}(t \mid x_{[0,t)}^0)} \right] \tag{18b}$$

This rearrangement is similar to that of equations (8a)–(8b). The intuition of equation (18a) is that we sample $M$ i.i.d. noise streams $x_{[0,T)}^1, \ldots, x_{[0,T)}^M$ for each possible real data $x_{[0,T)}^0$, sum up the log-ratio whenever $x^{0:M}$ has an event, and then take the expectation over all the possible real data $x_{[0,T)}^0$. The intuition of equation (18b) is that we draw noise samples $x_t^1, \ldots, x_t^M$ for each real

history $x^0_{[0,t)}$ at each time $t$, compute the log-ratio if $x^{0:M}_t$ has an event, take the expectation of the log-ratio over all the possible real histories and then sum over all the possible times. Therefore, these two expectations are equal.

We further rearrange equation (18) as

$$= \int_{t=0}^{T} \mathbb{E}_{x^0_{[0,t)} \sim p^*} \left[ \sum_{k=1}^{K} \left( \lambda^*_k(t \mid x^0_{[0,t)}) dt \log \frac{\lambda_k(t|x^0_{[0,t)})}{\underline{\lambda}_k(t|x^0_{[0,t)})} + \sum_{m=1}^{M} \lambda^q_k(t \mid x^0_{[0,t)}) dt \log \frac{\lambda^q_k(t|x^0_{[0,t)})}{\underline{\lambda}_k(t|x^0_{[0,t)})} \right) \right]$$
(19a)

$$= \int_{t=0}^{T} \mathbb{E}_{x^0_{[0,t)} \sim p^*} \left[ \sum_{k=1}^{K} \left( \lambda^*_k(t \mid x^0_{[0,t)}) dt \log \frac{\lambda_k(t|x^0_{[0,t)})}{\underline{\lambda}_k(t|x^0_{[0,t)})} + M \lambda^q_k(t \mid x^0_{[0,t)}) dt \log \frac{\lambda^q_k(t|x^0_{[0,t)})}{\underline{\lambda}_k(t|x^0_{[0,t)})} \right) \right]$$
(19b)

$$= \int_{t=0}^{T} \mathbb{E}_{x^0_{[0,t)} \sim p^*} \left[ \sum_{k=1}^{K} \underline{\lambda}^*_k(t \mid x^0_{[0,t)}) dt \left( \frac{\lambda^*_k(t|x^0_{[0,t)})}{\underline{\lambda}^*_k(t|x^0_{[0,t)})} \log \frac{\lambda_k(t|x^0_{[0,t)})}{\underline{\lambda}_k(t|x^0_{[0,t)})} + M \frac{\lambda^q_k(t|x^0_{[0,t)})}{\underline{\lambda}^*_k(t|x^0_{[0,t)})} \log \frac{\lambda^q_k(t|x^0_{[0,t)})}{\underline{\lambda}_k(t|x^0_{[0,t)})} \right) \right]$$
(19c)

where $\underline{\lambda}^*_k(t \mid x^0_{[0,t)}) \overset{\text{def}}{=} \lambda^*_k(t \mid x^0_{[0,t)}) + M \lambda^q_k(t \mid x^0_{[0,t)})$ can be thought of as the intensity of type $k$ under the superposition of $p^*$ and $M$ copies of $q$.

Now we obtain the final rearranged objective:

$$\int_{t=0}^{T} \sum_{x^0_{[0,t)}} p^*(x^0_{[0,t)}) \sum_{k=1}^{K} \underline{\lambda}^*_k(t \mid x^0_{[0,t)}) \underbrace{\left( \frac{\lambda^*_k(t|x^0_{[0,t)})}{\underline{\lambda}^*_k(t|x^0_{[0,t)})} \log \frac{\lambda_k(t|x^0_{[0,t)})}{\underline{\lambda}_k(t|x^0_{[0,t)})} + M \frac{\lambda^q_k(t|x^0_{[0,t)})}{\underline{\lambda}^*_k(t|x^0_{[0,t)})} \log \frac{\lambda^q_k(t|x^0_{[0,t)})}{\underline{\lambda}_k(t|x^0_{[0,t)})} \right)}_{\text{call it } H_\theta(k,t,x^0_{[0,t)})} dt$$
(20)

## B.2 Optimality Proof Details

In this section, we prove Theorem 1 that we stated in section 3.3. Recall the theorem:

**Theorem 1** (Optimality). *Under assumptions 1 and 2, $\theta \in \text{argmax}_\theta J_{\text{NC}}(\theta)$ if and only if $p_\theta = p^*$.*

We first need to highlight the key insight that $H_\theta(k, t, x^0_{[0,t)})$ in equation (20) is the negative cross-entropy between the following two discrete distributions over $\{\varnothing, 1, \ldots, K\}$:

$$\left[ \frac{\lambda^*_k(t|x^0_{[0,t)})}{\underline{\lambda}^*_k(t|x^0_{[0,t)})}, \frac{\lambda^q_k(t|x^0_{[0,t)})}{\underline{\lambda}^*_k(t|x^0_{[0,t)})}, \ldots, \frac{\lambda^q_k(t|x^0_{[0,t)})}{\underline{\lambda}^*_k(t|x^0_{[0,t)})} \right]$$
(21a)

$$\left[ \frac{\lambda_k(t|x^0_{[0,t)})}{\underline{\lambda}_k(t|x^0_{[0,t)})}, \underbrace{\frac{\lambda^q_k(t|x^0_{[0,t)})}{\underline{\lambda}_k(t|x^0_{[0,t)})}, \ldots, \frac{\lambda^q_k(t|x^0_{[0,t)})}{\underline{\lambda}_k(t|x^0_{[0,t)})}}_{\text{length is } M} \right]$$
(21b)

This negative cross-entropy is always smaller than or equal to the negative entropy of the distribution in equation (21a): it will be strictly smaller if these two distributions are distinct and equal when they are identical. Notice that in contrast to the negative cross-entropy at equation (9), this negative cross-entropy here is not infinitesimal.

*Proof.* The "if" part is straightforward to prove. Any $\theta$ for which $p_\theta = p^*$ would make $\lambda_k(t \mid x^0_{[0,t)}) = \lambda^*_k(t \mid x^0_{[0,t)})$, thus maximizing the negative cross-entropy between the two distributions in equation (21), for any type $k$ and any real history $x^0_{[0,t)}$ at any time $t$. Then the NCE objective in equation (20) is obviously maximized.

To check if any other $\bar{\theta} \notin \Theta^* \overset{\text{def}}{=} \{\theta^* : p_{\theta^*} = p^*\}$ maximizes $J_{\text{NC}}(\theta)$ as well, we analyze

$$J_{\text{NC}}(\bar{\theta}) - J_{\text{NC}}(\theta^*) = \int_{t=0}^{T} \sum_{x^0_{[0,t)}} p^*(x^0_{[0,t)}) \sum_{k=1}^{K} \underline{\lambda}^*_k(t \mid x^0_{[0,t)}) \underbrace{\left( H_{\bar{\theta}}(k, t, x^0_{[0,t)}) - H_{\theta^*}(k, t, x^0_{[0,t)}) \right)}_{\text{denote it as } G_{\bar{\theta}}(k,t,x^0_{[0,t)})} dt$$

where $\theta^*$ can be any member in $\Theta^*$. Note that $G_{\bar\theta}$ is not infinitesimal because the probabilities in $H$ and thus the entropy changes (if any) are not infinitesimal.

According to the definition of $\bar\theta$ and $\theta^*$, there must exist a stream $\bar x_{[0,T)}$, a time $\bar t \in (0,T)$ and a type $\bar k \in \{1,\ldots,K\}$ such that $\lambda_{\bar k}(\bar t \mid \bar x_{[0,\bar t)}) \neq \lambda_{\bar k}^*(\bar t \mid \bar x_{[0,\bar t)})$. Therefore, we have $G_{\bar\theta}(\bar k, \bar t, \bar x_{[0,\bar t)}) < 0$ since the distributions in equation (21) are distinct for the given history $\bar x_{[0,\bar t)}$. Does this difference lead to any overall change of the entire objective?

Actually, according to Lemma 1 in Appendix A, the existence of such $\bar x_{[0,T)}$, $\bar t$ and $\bar k$ implies that there exists an interval $(t', t'') \subset [0,T)$ such that, for any $t \in (t', t'')$, there exists a set $\mathcal{X}(t)$ of histories with non-zero measure such that any $x_{[0,t)} \in \mathcal{X}(t)$ satisfies $\lambda_{\bar k}(t \mid x_{[0,t)}) \neq \lambda_{\bar k}^*(t \mid x_{[0,t)})$. Then, given any of these histories, the entropy difference $G_{\bar\theta}$ would be $< 0$. That is to say, the following integral must be a non-infinitesimal negative number:

$$\int_{t=0}^{T} \sum_{x_{[0,t)}^0} p^*(x_{[0,t)}^0)\lambda_{\bar k}^*(t \mid x_{[0,t)}^0)G_{\bar\theta}(\bar k, t, x_{[0,t)}^0)dt \tag{22a}$$

$$= \int_{t=t'}^{t''} \sum_{x_{[0,t)}^0 \in \mathcal{X}(t)} p^*(x_{[0,t)}^0)\lambda_{\bar k}^*(t \mid x_{[0,t)}^0)G_{\bar\theta}(\bar k, t, x_{[0,t)}^0)dt \qquad (<0) \tag{22b}$$

$$+ \int_{t=t'}^{t''} \sum_{x_{[0,t)}^0 \notin \mathcal{X}(t)} p^*(x_{[0,t)}^0)\lambda_{\bar k}^*(t \mid x_{[0,t)}^0)G_{\bar\theta}(\bar k, t, x_{[0,t)}^0)dt \qquad (\leq 0) \tag{22c}$$

$$+ \int_{t=0}^{t'} \sum_{x_{[0,t)}^0} p^*(x_{[0,t)}^0)\lambda_{\bar k}^*(t \mid x_{[0,t)}^0)G_{\bar\theta}(\bar k, t, x_{[0,t)}^0)dt \qquad (\leq 0) \tag{22d}$$

$$+ \int_{t=t''}^{T} \sum_{x_{[0,t)}^0} p^*(x_{[0,t)}^0)\lambda_{\bar k}^*(t \mid x_{[0,t)}^0)G_{\bar\theta}(\bar k, t, x_{[0,t)}^0)dt \qquad (\leq 0) \tag{22e}$$

Therefore, the overall difference must be $< 0$ as well:

$$J_{\mathrm{LL}}(\bar\theta) - J_{\mathrm{LL}}(\theta^*) = \int_{t=0}^{T} \sum_{x_{[0,t)}^0} p^*(x_{[0,t)}^0) \sum_{k=1}^{K} \lambda_k^*(t \mid x_{[0,t)}^0)G_{\bar\theta}(k, t, x_{[0,t)}^0)dt \tag{23a}$$

$$= \int_{t=0}^{T} \sum_{x_{[0,t)}^0} p^*(x_{[0,t)}^0)\lambda_{\bar k}^*(t \mid x_{[0,t)}^0)G_{\bar\theta}(\bar k, t, x_{[0,t)}^0)dt \qquad (<0) \tag{23b}$$

$$+ \int_{t=0}^{T} \sum_{x_{[0,t)}^0} p^*(x_{[0,t)}^0) \sum_{k\neq\bar k} \lambda_k^*(t \mid x_{[0,t)}^0)G_{\bar\theta}(k, t, x_{[0,t)}^0)dt \qquad (\leq 0) \tag{23c}$$

Note that $J_{\mathrm{LL}}(\bar\theta) - J_{\mathrm{LL}}(\theta^*) < 0$ holds any $\bar\theta \notin \Theta^*$ and any $\theta^* \in \Theta^*$, meaning that $\theta^* \in \Theta^*$ is necessary to maximize the objective. Then the proof of the "only if" part is complete.

Now we have proved both the "if" and "only if" parts so the proof is complete.

$\qquad\qquad\qquad\qquad\qquad\qquad\qquad\qquad\qquad\qquad\qquad\qquad\qquad\qquad\qquad\qquad\qquad\qquad\quad \square$

### B.3  Consistency Proof Details

To discuss the statistical consistency (in this section) and efficiency (in Appendix B.4), we first need to spell out the empirical version of the objective

$$J_{\mathrm{NC}}^N(\theta) = \frac{1}{N}\sum_{n=1}^{N}\left(\sum_{t:x_{t,n}^0\neq\varnothing} \log\frac{\lambda_{x_{t,n}^0}(t\mid x_{[0,t),n}^0)}{\bar\lambda_{x_{t,n}^0}(t\mid x_{[0,t),n}^0)} + \sum_{m=1}^{M}\sum_{t:x_{t,n}^m\neq\varnothing} \log\frac{\lambda_{x_{t,n}^m}^{\mathrm q}(t\mid x_{[0,t),n}^0)}{\bar\lambda_{x_{t,n}^m}(t\mid x_{[0,t),n}^0)}\right) \tag{24}$$

where the subscript $n$ denotes the $n^{\mathrm{th}}$ i.i.d. draw of the observed sequence and the $M$ noise samples for this sequence. It is obvious that $\lim_{N\to\infty} J_{\mathrm{NC}}^N(\theta) \to J_{\mathrm{NC}}(\theta)$.

To analyze the consistency, we make the following assumptions:

**Assumption 3** (Continuity wrt. $\theta$). *For any history $x_{[0,t)}$ and event type $k \in \{1, \ldots, K\}$, $\lambda_k(t \mid x_{[0,t)})$ is continuous with respect to $\theta$.*

**Assumption 4** (Compactness). *The set of optimal parameters $\Theta^*$ is contained in the interior of a compact set $\Theta \subset \mathbb{R}^{|\theta|}$.*

They are analogous to assumptions 4.2 and 4.3 of Ma & Collins (2018) respectively.

Our NCE method turns out to be strongly consistent in the sense that:

**Theorem 2** (Consistency). *Under assumptions 2, 3 and 4, for any $\theta \in \Theta_{\mathrm{NC}}^N \overset{\mathrm{def}}{=} \mathrm{argmax}_\theta J_{\mathrm{NC}}^N(\theta)$ and $M \geq 1$, with probability 1, we have $\lim_{N\to\infty} \min_{\theta^* \in \Theta^*} \|\theta - \theta^*\| = 0$ where $\|\cdot\|$ is the $L_2$ norm.*

The intuition of this theorem is that: since the two functions $J_{\mathrm{NC}}^N(\theta)$ and $J_{\mathrm{NC}}(\theta)$ will become the same as $N \to \infty$ and they are continuous with respect to $\theta$, then any $\theta \in \mathrm{argmax}_\theta J_{\mathrm{NC}}^N(\theta)$ has to be close to some member of the set $\mathrm{argmax}_\theta J_{\mathrm{NC}}(\theta)$. The full proof is almost identical to the proof of Theorem 4.2 in Ma & Collins (2018). But we will still spell it out in our notation for completeness.

*Proof.* Under the assumption in Theorem 2, by classical large sample theory (Ferguson, 1996), we have

$$\mathbb{P}\left[ \lim_{N\to\infty} \sup_{\theta\in\Theta'} |J_{\mathrm{NC}}^N(\theta) - J_{\mathrm{NC}}(\theta)| = 0 \right] = 1 \text{ for any compact set } \Theta' \subset \Theta \tag{25}$$

where $\mathbb{P}$ stands for "probability". Since $|J_{\mathrm{NC}}^N(\theta) - J_{\mathrm{NC}}(\theta)| \geq J_{\mathrm{NC}}^N(\theta) - J_{\mathrm{NC}}(\theta)$, we have

$$\mathbb{P}\left[ \limsup_{N\to\infty} \sup_{\theta\in\Theta'} (J_{\mathrm{NC}}^N(\theta) - J_{\mathrm{NC}}(\theta)) \leq 0 \right] = 1 \tag{26}$$

Moreover, for any $\theta'^N \in \mathrm{argmax}_{\theta\in\Theta'} J_{\mathrm{NC}}^N(\theta)$, we have

$$\sup_{\theta\in\Theta'} (J_{\mathrm{NC}}^N(\theta) - J_{\mathrm{NC}}(\theta)) \geq J_{\mathrm{NC}}^N(\theta'^N) - J_{\mathrm{NC}}(\theta'^N) \geq \sup_{\theta\in\Theta'} J_{\mathrm{NC}}^N(\theta) - \sup_{\theta\in\Theta'} J_{\mathrm{NC}}(\theta) \tag{27}$$

Plugging equation (27) into equation (26) gives

$$\mathbb{P}\left[ \limsup_{N\to\infty} \sup_{\theta\in\Theta'} J_{\mathrm{NC}}^N(\theta) - \sup_{\theta\in\Theta'} J_{\mathrm{NC}}(\theta) \leq 0 \right] = \mathbb{P}\left[ \limsup_{N\to\infty} \sup_{\theta\in\Theta'} J_{\mathrm{NC}}^N(\theta) \leq \sup_{\theta\in\Theta'} J_{\mathrm{NC}}(\theta) \right] = 1 \tag{28}$$

For any $\delta > 0$, we define $\Theta_\delta \overset{\mathrm{def}}{=} \{\theta : \min_{\theta^*\in\Theta^*} \|\theta - \theta^*\| > \delta\}$ and have

$$\mathbb{P}\left[ \limsup_{N\to\infty} \sup_{\theta\in\Theta_\delta} J_{\mathrm{NC}}^N(\theta) \leq \sup_{\theta\in\Theta_\delta} J_{\mathrm{NC}}(\theta) < \sup_{\theta\in\Theta} J_{\mathrm{NC}}(\theta) \right] = 1 \tag{29}$$

On the other hand, we also have $|J_{\mathrm{NC}}^N(\theta) - J_{\mathrm{NC}}(\theta)| \geq J_{\mathrm{NC}}(\theta) - J_{\mathrm{NC}}^N(\theta)$, which gives

$$\mathbb{P}\left[ \limsup_{N\to\infty} \sup_{\theta\in\Theta'} (J_{\mathrm{NC}}(\theta) - J_{\mathrm{NC}}^N(\theta)) \leq 0 \right] = 1 \tag{30}$$

For any $\theta' \in \mathrm{argmax}_{\theta\in\Theta'} J_{\mathrm{NC}}(\theta)$, we have

$$\sup_{\theta\in\Theta'} (J_{\mathrm{NC}}(\theta) - J_{\mathrm{NC}}^N(\theta)) \geq J_{\mathrm{NC}}(\theta') - J_{\mathrm{NC}}^N(\theta') \geq \sup_{\theta\in\Theta'} J_{\mathrm{NC}}(\theta) - \sup_{\theta\in\Theta'} J_{\mathrm{NC}}^N(\theta) \tag{31}$$

Plugging equation (31) into equation (30) gives

$$\mathbb{P}\left[ \sup_{\theta\in\Theta'} J_{\mathrm{NC}}(\theta) + \limsup_{N\to\infty}(- \sup_{\theta\in\Theta'} J_{\mathrm{NC}}^N(\theta)) \leq 0 \right] = \mathbb{P}\left[ \liminf_{N\to\infty} \sup_{\theta\in\Theta'} J_{\mathrm{NC}}^N(\theta) \geq \sup_{\theta\in\Theta'} J_{\mathrm{NC}}(\theta) \right] = 1 \tag{32}$$

which, when we let $\Theta' = \Theta$, gives

$$\mathbb{P}\left[ \liminf_{N\to\infty} \sup_{\theta\in\Theta} J_{\mathrm{NC}}^N(\theta) \geq \sup_{\theta\in\Theta} J_{\mathrm{NC}}(\theta) \right] = 1 \tag{33}$$

Combining equation (29) and equation (33), we have that, for any $\theta^N \in \Theta^N \overset{\mathrm{def}}{=} \mathrm{argmax}_\theta J_{\mathrm{NC}}^N(\theta)$ (defined in Theorem 2), there exists an integer $N'$ such that for any $N \geq N'$

$$\mathbb{P}\left[ \theta^N \notin \Theta_\delta \right] = 1 \tag{34}$$

which holds for any $\delta > 0$ and thus gives

$$\mathbb{P}\left[ \lim_{N\to\infty} \min_{\theta^*\in\Theta^*} \|\theta^N - \theta^*\| = 0 \right] = 1 \tag{35}$$

which completes the proof of Theorem 2.

$\square$

## B.4 Efficiency Proof Details

To quantify the statistical efficiency of our method, we make the following assumptions:

**Assumption 5** (Identifiability). *There is only one parameter vector $\theta^*$ such that $p_{\theta^*} = p^*$.*

**Assumption 6** (Differentiability). *For any history $x_{[0,t)}$ and event type $k \in \{1, \ldots, K\}$, $\lambda_k(t \mid x_{[0,t)})$ is twice continuously differentiable with respect to $\theta$.*

**Assumption 7** (Singularity). *The Fisher information matrix $\mathbf{I}_*$ under the model $p_\theta$ is non-singular.*

They are analogous to assumptions 4.4, 4.6 and 4.7 of Ma & Collins (2018) respectively.

Before we show the efficiency of our method, we first spell out the definition of $\mathbf{I}_*$:

$$\mathbf{I}_* \overset{\text{def}}{=} \mathbb{E}_{x_{[0,T)} \sim p^*} \left[ \nabla_\theta \log p_{\theta^*}(x_{[0,T)}) \nabla_\theta \log p_{\theta^*}(x_{[0,T)})^\top \right] \tag{36}$$

where $\nabla_\theta \log p_{\theta^*}$ stands for "the gradient of $\log p_\theta$ with respect to $\theta$ at $\theta = \theta^*$." This formula can be rearranged as

$$\int_{t=0}^{T} \mathbb{E}_{x_{[0,t)} \sim p^*} \left[ \mathbb{E}_{x_{[t,t+dt)} \sim p^*} \left[ \nabla_\theta \log p_{\theta^*}(x_{[t,t+dt)} \mid x_{[0,t)}) \nabla_\theta \log p_{\theta^*}(x_{[t,t+dt)} \mid x_{[0,t)})^\top \right] \right] \tag{37a}$$

$$= \int_{t=0}^{T} \mathbb{E}_{x_{[0,t)} \sim p^*} \left[ \mathbb{E}_{x_{[t,t+dt)} \sim p^*} \left[ \frac{\nabla_\theta p_{\theta^*}(x_{[t,t+dt)} \mid x_{[0,t)})}{p_{\theta^*}(x_{[t,t+dt)} \mid x_{[0,t)})} \frac{\nabla_\theta p_{\theta^*}(x_{[t,t+dt)} \mid x_{[0,t)})^\top}{p_{\theta^*}(x_{[t,t+dt)} \mid x_{[0,t)})} \right] \right] \tag{37b}$$

$$= \int_{t=0}^{T} \mathbb{E}_{x_{[0,t)} \sim p^*} \left[ \sum_{x_{[t,t+dt)}} \frac{\nabla_\theta p_{\theta^*}(x_{[t,t+dt)} \mid x_{[0,t)}) \nabla_\theta p_{\theta^*}(x_{[t,t+dt)} \mid x_{[0,t)})^\top}{p_{\theta^*}(x_{[t,t+dt)} \mid x_{[0,t)})} \right] \tag{37c}$$

Technically, $x_{[t,t+dt)}$ will have an event of type $k$ with probability $\lambda_k^*(t)dt$ under $p^*$ ($\lambda_k(t)dt$ under $p_\theta$) or has no event at all with probability $1 - \sum_{k=1}^{K} \lambda_k^*(t)dt$ under $p^*$ ($1 - \sum_{k=1}^{K} \lambda_k(t)dt$ under $p_\theta$). In the former case, we have $\nabla_\theta p_{\theta^*} \nabla_\theta p_{\theta^*}^\top / p_{\theta^*} = \nabla_\theta \lambda_k^*(t) \nabla_\theta \lambda_k^*(t)^\top dt / \lambda_k^*(t)$; in the latter case, we have $\nabla_\theta p_{\theta^*} = -\sum_{k=1}^{K} \nabla_\theta \lambda_k^*(t)dt$ but $p_{\theta^*} \approx 1$, so $\nabla_\theta p_{\theta^*} \nabla_\theta p_{\theta^*}^\top / p_{\theta^*} = o(dt)$ can be ignored. Plugging these quantities into equation (37) gives us

$$\mathbf{I}_* = \int_{t=0}^{T} \mathbb{E}_{x_{[0,t)} \sim p^*} \left[ \sum_{k=1}^{K} \frac{\nabla_\theta \lambda_k^*(t \mid x_{[0,t)}) \nabla_\theta \lambda_k^*(t \mid x_{[0,t)})^\top}{\lambda_k^*(t \mid x_{[0,t)})} dt \right] \tag{38a}$$

$$= \int_{t=0}^{T} \sum_{x_{[0,t)}} p^*(x_{[0,t)}) \sum_{k=1}^{K} \frac{\nabla_\theta \lambda_k^*(t \mid x_{[0,t)}) \nabla_\theta \lambda_k^*(t \mid x_{[0,t)})^\top}{\lambda_k^*(t \mid x_{[0,t)})} dt \tag{38b}$$

Note that $\nabla_\theta \lambda_k^*(t)$ stands for "the gradient of $\lambda_k(t)$ with respect to $\theta$ at $\theta = \theta^*$."

Now we proceed to our efficiency theorem. We denote the unique optimal parameter vector as $\theta^*$ and use $\hat{\theta}$ for the estimate given by maximizing $J_{\text{NC}}^N(\theta)$. It turns out that our method approaches *Fisher efficiency* as $M$ grows.

**Theorem 3** (Efficiency). *Under assumptions 2 and 4–7, there exists an integer $\bar{M}$ such that for all $M > \bar{M}$*

$$\sqrt{N}(\hat{\theta} - \theta^*) \to \text{Normal}(0, \mathbf{I}_M^{-1}) \text{ as } N \to \infty \tag{39}$$

*for some non-singular matrix $\mathbf{I}_M^{-1}$. Moreover, there exist a constant $C > 0$ such that for all $M > \bar{M}$*

$$\|\mathbf{I}_M^{-1} - \mathbf{I}_*^{-1}\| \leq C/M \tag{40}$$

*where $\|\mathbf{I}\|$ is the spectral norm of matrix $\mathbf{I}$.*

*Proof.* We first prove that $\sqrt{N}(\hat{\theta} - \theta^*)$ is asymptotically normal. By the Mean-Value Theorem, we have

$$\nabla_\theta J_{\text{NC}}^N(\hat{\theta}) = \nabla_\theta J_{\text{NC}}^N(\theta^*) + (\hat{\theta} - \theta^*) \int_{u=0}^{1} \nabla_\theta^2 J_{\text{NC}}^N(\theta^* + u(\hat{\theta} - \theta^*)) dt \tag{41}$$

Since $\hat{\theta}$ maximizes $J_{\text{NC}}^N$, we have

$$\hat{\theta} - \theta^* = \left[ -\int_{u=0}^1 \nabla_\theta^2 J_{\text{NC}}^N(\theta^* + u(\hat{\theta} - \theta^*))dt \right]^{-1} \nabla_\theta J_{\text{NC}}^N(\theta^*) \tag{42}$$

By Law of Large Numbers and Theorem 2, we have

$$\int_{u=0}^1 \nabla_\theta^2 J_{\text{NC}}^N(\theta^* + u(\hat{\theta} - \theta^*))dt \to \underbrace{\mathbb{E}_{x_{[0,T)}^0 \sim p^*, x_{[0,T)}^{1:M} \sim q} \left[ \nabla_\theta^2 L(\theta^*) \right]}_{\text{short as } \mathbb{E}\left[ \nabla_\theta^2 L(\theta^*) \right]} \text{ as } N \to \infty \tag{43}$$

where $L(\theta)$ is defined as the objective for a random draw of $x_{[0,T)}^{0:M}$ and thus is just the term inside the expectation of equation (6):

$$L(\theta) \stackrel{\text{def}}{=} \sum_{t:x_t^0 \neq \varnothing} \log \frac{\lambda_{x_t^0}(t|x_{[0,t)}^0)}{\underline{\lambda}_{x_t^0}(t|x_{[0,t)}^0)} + \sum_{m=1}^M \sum_{t:x_t^m \neq \varnothing} \log \frac{\lambda_{x_t^m}^{\text{q}}(t|x_{[0,t)}^0)}{\underline{\lambda}_{x_t^m}(t|x_{[0,t)}^0)} \tag{44}$$

The term $\nabla_\theta^2 L(\theta^*)$ stands for "the Hessian matrix of $L(\theta)$ with respect to $\theta$ at $\theta = \theta^*$." As for $\nabla_\theta J_{\text{NC}}^N(\theta^*)$, by Central Limit Theorem, we have

$$\sqrt{N} \nabla_\theta J_{\text{NC}}^N(\theta^*) \to \text{Normal}(0, \underbrace{\mathbb{E}_{x_{[0,T)}^0 \sim p^*, x_{[0,T)}^{1:M} \sim q} \left[ \nabla_\theta L(\theta^*) \nabla_\theta L(\theta^*)^\top \right]}_{\text{short as } \mathbb{V}[\nabla_\theta L(\theta^*)]}) \tag{45}$$

Combining equations (42), (43) and (45), we obtain the asymptotic normality

$$\sqrt{N}(\hat{\theta} - \theta^*) \to \text{Normal}(0, \mathbb{E}\left[ \nabla_\theta^2 L(\theta^*) \right]^{-1} \mathbb{V}[\nabla_\theta L(\theta^*)] \mathbb{E}\left[ \nabla_\theta^2 L(\theta^*) \right]^{-1}) \tag{46}$$

Now we compute the covariance matrix of the asymptotic normal distribution. Following steps similar to equations (18) and (19), we rearrange $\mathbb{E}\left[ \nabla_\theta^2 L(\theta^*) \right]$ to be

$$\mathbb{E}\left[ \nabla_\theta^2 L(\theta^*) \right] = \int_{t=0}^T \mathbb{E}_{x_{[0,t)}^0 \sim p^*} \left[ \sum_{k=1}^K \left( \lambda_k^*(t)dt \nabla_\theta^2 \log \frac{\lambda_k^*(t)}{\underline{\lambda}_k^*(t)} + M\lambda_k^{\text{q}}(t)dt \nabla_\theta^2 \log \frac{\lambda_k^{\text{q}}(t)}{\underline{\lambda}_k^*(t)} \right) \right] \tag{47a}$$

$$= \int_{t=0}^T \mathbb{E}_{x_{[0,t)}^0 \sim p^*} \left[ \sum_{k=1}^K (\frac{1}{\underline{\lambda}_k^*(t)} - \frac{1}{\lambda_k^*(t)}) \nabla_\theta \lambda_k^*(t) \nabla_\theta \lambda_k^*(t)^\top dt \right] \tag{47b}$$

$$= \int_{t=0}^T p^*(x_{[0,t)}^0) \sum_{k=1}^K (\frac{1}{\underline{\lambda}_k^*(t)} - \frac{1}{\lambda_k^*(t)}) \nabla_\theta \lambda_k^*(t) \nabla_\theta \lambda_k^*(t)^\top dt \tag{47c}$$

where we omit the condition $x_{[0,t)}^0$ in the probabilities and intensities for presentation simplicity. We also omit the tedious arithmetic manipulation that spells $\nabla_\theta^2 \log(\lambda/\underline{\lambda})$ out.

Following similar steps, we then rearrange $\mathbb{V}[\nabla_\theta L(\theta^*)]$ to be

$$\int_{t=0}^T \mathbb{E}_{x_{[0,t)}^0 \sim p^*} \left[ \sum_{k=1}^K \left( \lambda_k^*(t)dt \nabla_\theta \nabla_\theta^\top \log \frac{\lambda_k^*(t)}{\underline{\lambda}_k^*(t)} + M\lambda_k^{\text{q}}(t)dt \nabla_\theta \nabla_\theta^\top \log \frac{\lambda_k^{\text{q}}(t)}{\underline{\lambda}_k^*(t)} \right) \right] \tag{48a}$$

$$= \int_{t=0}^T \mathbb{E}_{x_{[0,t)}^0 \sim p^*} \left[ \sum_{k=1}^K (\frac{1}{\lambda_k^*(t)} - \frac{1}{\underline{\lambda}_k^*(t)}) \nabla_\theta \lambda_k^*(t) \nabla_\theta \lambda_k^*(t)^\top dt \right] \tag{48b}$$

$$= \mathbb{E}\left[ -\nabla_\theta^2 L(\theta^*) \right] \tag{48c}$$

where we use $\nabla_\theta \nabla_\theta^\top f(\theta)$ to denote $(\nabla_\theta f(\theta))(\nabla_\theta f(\theta))^\top$. For presentation simplicity, we omit the arithmetic manipulation that spells $\nabla_\theta \nabla_\theta^\top \log(\lambda/\underline{\lambda})$ out.

Then we can simplify the asymptotic normality to be

$$\sqrt{N}(\hat{\theta} - \theta^*) \to \text{Normal}(0, \mathbb{E}\left[ -\nabla_\theta^2 L(\theta^*) \right]^{-1}) \tag{49}$$

We can think of $\mathbf{I}_M \overset{\text{def}}{=} \mathbb{E}\left[-\nabla_\theta^2 L(\theta^*)\right]$ as the "information matrix" of our objective $J_{\text{NC}}(\theta)$. And its relation with the Fisher information matrix $\mathbf{I}_*$ is:

$$\mathbf{I}_M = \mathbf{I}_* - \underbrace{\int_{t=0}^{T} \sum_{x_{[0,t)}^0} p^*(x_{[0,t)}^0) \sum_{k=1}^{K} \frac{1}{\lambda_k^*(t) + M\lambda_k^{\text{q}}(t)} \nabla_\theta \lambda_k^*(t) \nabla_\theta \lambda_k^*(t)^\top \, dt}_{\text{call it } \Delta\mathbf{I}} \tag{50}$$

Apparently, when $M$ is large enough, $\mathbf{I}_M$ will be non-singular. Precisely, since $\mathbf{I}_*$ is non-singular, there must exist $\bar{M} > 0$ such that, for any $M > \bar{M}$, $0 < \|\Delta\mathbf{I}\| \leq \sigma(\mathbf{I}_*)/2$ where $\sigma(\mathbf{I})$ is the *smallest* singular value of matrix $\mathbf{I}$ and $\|\mathbf{I}\|$ is the *spectral norm*, i.e., the *largest* singular value, of matrix $\mathbf{I}$. By Weyl's inequality, we have $\sigma(\mathbf{I}_M) \geq \sigma(\mathbf{I}_*) - \|\Delta\mathbf{I}\| \geq \sigma(\mathbf{I}_*)/2$, meaning that $\mathbf{I}_M$ is non-singular.

Now we can start analyzing $\|\mathbf{I}_M^{-1} - \mathbf{I}_*^{-1}\|$. By the definition of the spectral norm, we have:

$$\|\mathbf{I}_M^{-1} - \mathbf{I}_*^{-1}\| = \|\mathbf{I}_*^{-1}(\mathbf{I}_* - \mathbf{I}_M)\mathbf{I}_M^{-1}\| \leq \|\mathbf{I}_*^{-1}\|\|\Delta\mathbf{I}\|\|\mathbf{I}_M^{-1}\| \leq \frac{1}{\sigma(\mathbf{I}_*)}\|\Delta\mathbf{I}\|\frac{2}{\sigma(\mathbf{I}_*)} \tag{51}$$

Since the intensity functions are all bounded, continuous and twice continuously differentiable, $\|\nabla_\theta \lambda_k^*(t) \nabla_\theta \lambda_k^*(t)^\top\|$ will be bounded, meaning that $\|\Delta\mathbf{I}\|$ will be bounded as well. Moreover, the ratio $\lambda_k^*(t)/\lambda_k^{\text{q}}(t)$ is also bounded. We define $r = \sup_{x_{[0,t)}^0, k} \frac{\lambda_k^*(t|x_{[0,t)}^0)}{\lambda_k^{\text{q}}(t|x_{[0,t)}^0)}$ and have $M\lambda_k^{\text{q}}(t) \geq M\lambda_k^*(t)/r$. Then there must exist $B > 0$ such that we have:

$$\|(1 + \tfrac{M}{r})\Delta\mathbf{I}\| \leq B\|\mathbf{I}_*\| \Rightarrow \|\Delta\mathbf{I}\| \leq \tfrac{rB}{r+M}\|\mathbf{I}_*\| < \tfrac{1}{M}rB\|\mathbf{I}_*\| \tag{52}$$

Combining equations (51) and (52), we have

$$\|\mathbf{I}_M^{-1} - \mathbf{I}_*^{-1}\| \leq \tfrac{1}{M}\underbrace{\tfrac{2}{\sigma(\mathbf{I}_*)^2}rB\|\mathbf{I}_*\|}_{\text{call it } C} \tag{53}$$

meaning that there exists $C > 0$ such that, for any $M > \bar{M}$, $\|\mathbf{I}_M^{-1} - \mathbf{I}_*^{-1}\| \leq C/M$.

Note that the ratio $r$ reflects the effect of $\lambda_k^{\text{q}}(t)$ on the efficiency. In the special case of $q = p^*$, we have $r = 1$ and $\Delta\mathbf{I} = \frac{1}{M+1}\mathbf{I}_*$ and the asymptotic covariance matrix becomes $(1 + \frac{1}{M})\mathbf{I}_*^{-1}$.

This completes our proof.

$\square$

## C  Algorithm Details

### C.1  NCE Objective Computation Details

Our main algorithm is presented as Algorithm 1. It covers the recipe for computing our NCE objective, as well as the algorithm to sample from $q$.

### C.2  Training the Noise Distribution $q$ by NCE

Before we optimize our $J_{\text{NC}}(\theta)$, we first fit the noise distribution $q$ to the training data. As discussed in endnote 7, we expect that fitting the data well will give a good training signal to learn $\theta$.

In the experiments of this paper, we used MLE to estimate the parameters $\phi$ of $q$, which involves taking approximate integrals as in Mei & Eisner (2017). (After all, we did not yet know whether NCE would work well.) To avoid the approximate integrals, however, one could instead estimate $\phi$ using NCE. When evaluating this NCE objective during training of $\phi$, one can take the noise distribution to be $q_{\phi_{\text{old}}}$ where $\phi_{\text{old}}$ is any snapshot of $\phi$ from a recent iteration of training (even the current iteration). The same $\phi_{\text{old}}$ must be used for both drawing noise events via the thinning algorithm, and for scoring these noise events and their contrasting observed events.

Regardless of whether we use MLE or NCE, it is faster to train $q$ than to train $p$ because $q$ only has $C$ event types instead of $K$.

The idea of using as the noise distribution a model previously trained with NCE was also considered in the original NCE paper (Gutmann & Hyvärinen, 2010).

**Algorithm 1** Training Objective Computation for Noise-Contrastive Estimation.

---

**Input:** observed event stream $x_{[0,T)}$ with $I$ events at times $0 = t_0 < t_1 < \ldots t_I < t_{I+1} = T$; model $p_\theta$; noise distribution $q$; number of noise samples $M$

**Output:** training objective $J_{\mathrm{NC}}$ evaluated on $x_{[0,T)}$ and the corresponding noise samples

1: **procedure** COMPUTEOBJECTIVE($\mathbf{x}, p_\theta, q, M$)
2:    ▷ *algorithm input $p_\theta$ gives info to define intensity function $\lambda_k(t)$*
3:    $J_{\mathrm{NC}} \leftarrow 0$                                            ▷ *initialize the objective*
4:    initialize the neural states $s$ and $s^{\mathrm{q}}$ of $p_\theta$ and $q$ respectively      ▷ *i.e., their LSTM states*
5:    $i \leftarrow 0$
6:    **while** $i \leq I$ :
7:      $i \mathrel{+}= 1$
8:      ▷ *use noise samples in the current interval*
9:      **for** $(t, k, \lambda^{\mathrm{q}}, \mu)$ **in** DRAWNOISESAMPLES($t_{i-1}, t_i$) :
10:        compute the model intensity $\lambda_k(t \mid s)$ under $p_\theta$
11:        $J_{\mathrm{NC}} \mathrel{+}= \mu \log \frac{\lambda^{\mathrm{q}}}{\lambda_k(t\mid s) + M\lambda^{\mathrm{q}}}$
12:      **if** $i > I$ : **break**
13:      ▷ *use the real event at time $t_i$*
14:      $t \leftarrow t_i, k \leftarrow x_{t_i}$
15:      compute the model intensity $\lambda_k(t \mid s)$ under $p_\theta$
16:      compute the noise intensity $\lambda_k^{\mathrm{q}}(t \mid s^{\mathrm{q}})$ under $q$
17:      $J_{\mathrm{NC}} \mathrel{+}= \log \frac{\lambda_k(t\mid s)}{\lambda_k(t\mid s) + M\lambda_k^{\mathrm{q}}(t\mid s^{\mathrm{q}})}$
18:      update the neural states $s$ and $s^{\mathrm{q}}$ of $p_\theta$ and $q$ respectively with this real event
19:    **return** $J_{\mathrm{NC}}$
20: **procedure** DRAWNOISESAMPLES($t_{\mathrm{beg}}, t_{\mathrm{end}}$)       ▷ *draw noise samples over interval $(t_{beg}, t_{end})$*
21:    ▷ *has access to $q, M$*
22:    ▷ *define the **total intensity function** $\lambda^{\mathrm{q}}(t \mid s^{\mathrm{q}}) \stackrel{\text{def}}{=} \sum_{c=1}^{C} \lambda_c^{\mathrm{q}}(t \mid s^{\mathrm{q}})$*
23:    $\mathcal{Q} \leftarrow$ empty collection                          ▷ *collection of noise samples*
24:    $t \leftarrow t_{\mathrm{beg}}$; find any $\overline{\lambda} \geq \sup \{\lambda^{\mathrm{q}}(t \mid s^{\mathrm{q}}) : t \in (t_{\mathrm{beg}}, t_{\mathrm{end}})\}$
25:    **repeat**
26:      draw $\Delta \sim \mathrm{Exp}(M\overline{\lambda})$; $t \mathrel{+}= \Delta$             ▷ *propose a noise time*
27:      **if** $t < t_{\mathrm{end}}$ :
28:        $\mu \leftarrow \lambda^{\mathrm{q}}(t \mid s^{\mathrm{q}})/\overline{\lambda}$        ▷ *compute probability to accept the proposed time*
29:        **if** $\mu < 0.05$ :              ▷ *stochastically accept $t$ with prob $\mu$ if $\mu < 0.05$*
30:          $u \sim \mathrm{Unif}(0,1)$; **if** $u < \mu : \mu \leftarrow 1$
31:        **if** $\mu \geq 0.05$ :            ▷ *otherwise fractionally accept $t$ with weight $\mu$*
32:          draw $c \in \{1, \ldots, C\}$ where probability of $c$ is $\propto \lambda_c^{\mathrm{q}}(t \mid s^{\mathrm{q}})$    ▷ *choose coarse type*
33:          draw $k \in \{1, \ldots, K\}$ where probability of $k$ is $q(k \mid c)$    ▷ *choose refinement*
34:          compute the noise intensity $\lambda_k^{\mathrm{q}}(t \mid s^{\mathrm{q}})$ under $q$
35:          add $(t, k, \lambda_k^{\mathrm{q}}(t \mid s^{\mathrm{q}}), \mu)$ to $\mathcal{Q}$
36:    **until** $t \geq t_{\mathrm{end}}$
37:    **return** $\mathcal{Q}$

---

| DATASET | $K$ | # OF EVENT TOKENS | | | SEQUENCE LENGTH | | |
|---|---|---|---|---|---|---|---|
| | | TRAIN | DEV | TEST | MIN | MEAN | MAX |
| SYNTHETIC-1 | 10000 | 100000 | 10000 | 10000 | 100 | 100 | 100 |
| SYNTHETIC-2 | 10000 | 100000 | 10000 | 10000 | 100 | 100 | 100 |
| EUROEMAIL | 10000 | 50000 | 10000 | 10000 | 100 | 100 | 100 |
| BITCOINOTC | 19800 | 1000 | 500 | 500 | 100 | 100 | 100 |
| COLLEGEMSG | 9900 | 8000 | 1000 | 1000 | 100 | 100 | 100 |
| WIKITALK | 10000 | 100000 | 20000 | 20000 | 100 | 100 | 100 |
| ROBOCUP | 528 | 2195 | 817 | 780 | 780 | 948 | 1336 |
| IPTV | 49000 | 27355 | 4409 | 4838 | 36602 | 36602 | 36602 |

Table 1: Statistics of each dataset. For IPTV, we have a single long sequence of 36602 tokens: we use the first 27355 as training data, the next 4409 as dev data and the remaining 4838 as test data. For other datasets, training, dev and test sequences are separate sequences.

# D    Experimental Details and Additional Results

## D.1    Dataset Details

Besides the datasets we have introduced in section 5, we also run experiments on the following real-world social interaction datasets:

**CollegeMsg** (Panzarasa et al., 2009).   This dataset contains anonymized private messages sent on an online social network at an university. Each record $(u, v, t)$ means that user $u$ sent a private message to user $v$ at time $t$ and each $u, v$ pair is an event type. We consider the top 100 users sorted by the number of messages they sent and received: the total number of possible event types is then $K = 9900$ since self-messaging is not allowed.

**WikiTalk** (Leskovec et al., 2010).   This dataset contains the records of anonymized Wikipedia users editing each other's Talk page. Each record $(u, v, t)$ means that user $u$ edited user $v$'s talk page at time $t$ and each $u, v$ pair is an event type. We consider the top 100 users sorted by the number of edits they made and received and the total number of possible event types is $K = 10000$.

Table 1 shows statistics about each dataset that we use in this paper.

## D.2    Training Details

For each of the chosen models in section 5, the only hyperparameter to tune is the hidden dimension $D$ of the neural network. On each dataset, we searched for $D$ that achieves the best performance on the dev set. Our search space is $\{4, 8, 16, 32, 64, 128\}$.

For learning, we used the Adam algorithm (Kingma & Ba, 2015) with its default settings. For each $\rho$ or $M$, we run training long enough so that the log-likelihood on the held-out data can converge.

## D.3    More Results on Real-World Social Interaction Datasets

The learning curves on CollegeMsg and WikiTalk datasets are shown in Figure 4: they look similar to those in Figure 2 and lead to the same conclusions.

## D.4    Ablation Study I: Always or Never Redraw Noise samples

In Figure 5, we show the learning curves for the "always redraw" and "never redraw" strategies on the first synthetic dataset. As shown in Figure 5a, with the "always redraw" strategy, NCE (━━) needs considerably fewer intensity evaluations to reach the highest log-likelihood (▪▪▪) that MLE (━━) can achieve on the held-out data. However, the curve with $M = 1000$ increases more slowly than MLE in terms of wall-clock time since it spends too much time on drawing new noise samples.

As shown in Figure 5b, with the "never redraw" strategy, $M = 1000$ overtakes MLE: a single draw of $M = 1000$ noise streams is able to give very good training signals and the saved computation can be spent on training $p_\theta$ repeatedly on the same samples. However, the curve of $M = 1$ only achieves log-likelihood $\approx -200$ and thus falls out of the zoomed-in view.

(a1) CollegeMsg: neural $q$              (a2) CollegeMsg: Poisson $q$

(b1) WikiTalk: neural $q$              (b2) WikiTalk: Poisson $q$

Figure 4: Learning curves of MLE and NCE on the other real-world social interaction datasets.

(a) Always redraw new noise samples        (b) Never redraw new noise samples

Figure 5: Ablation Study I. Learning curves of MLE and NCE with $q = p^*$ and different "redraw" strategies.

(a) EuroEmail                  (b) BitcoinOTC

(c) CollegeMsg                  (d) WikiTalk

Figure 6: Ablation Study II. Learning curves of MLE and NCE with untrained $q$ on social interaction datasets.

(a) RoboCup                    (b) IPTV

Figure 7: Ablation Study III. Learning curves of MLE and NCE using neural $q$ with $C = 1$.

**D.5   Ablation Study II: NCE with Untrained Noise Distribution**

In Figure 6, we show the learning curves of NCE with untrained noise distributions on the real-world social interaction datasets. As we can see, NCE in this setting tends to end up with worse generalization (interestingly except on WikiTalk) and suffers slow convergence (on BitcoinOTC and CollegeMsg) and large variance (on BitcoinOTC).

**D.6   Ablation Study III: Effect of $C$**

In Figure 7, we show learning curves of NCE using the neural $q$ with $C = 1$. Taking $C = 1$ means that the same number of noise samples can be drawn faster (with fewer intensity evaluations). However, more training epochs may be needed because the noise looks less like true observations and so NCE's discrimination tasks are less challenging (see endnote 7).

On the RoboCup dataset, $C = 1$ exhibits similar learning speed to $C = 5$ but has slightly worse generalization. On the IPTV dataset, $C = 1$ gives a considerable speedup over $C = 49$ without harming the final generalization. The NCE curves for $M = 5$ and $M = 10$ shift substantially to the left, since $C = 1$ requires *many* fewer intensity evaluations.