[Reviews · NeurIPS 2020]

Review 1

Summary and Contributions: This paper proposes a version of noise-contrastive estimation (NCE) method to alleviate computational cost for multivariate point processes and provides its theoretical guarantees. The authors evaluate their work on both synthetic and real-world datasets and show that their method achieve comparable results with much less computational time compared with baselines. However, the assumptions shown in the theoretical part seems to mismatch with the experimental results.

Strengths: Applying NCE to make the learning of point process scalable is a very good idea. Moreover, the authors provide theoretical support on the rationality of the proposed learning strategy, which improves the solidness of the proposed method. The proof seems correct.

Weaknesses: The main concern is the experimental part. Although the training/testing likelihood is reasonable for evaluating the convergence and the performance of the proposed method, I would like to see more comparisons on predictive tasks in real-world data sets. Additionally, the assumption 1 in the paper may be questionable in some situations. The continuity is a strong assumption on the intensity function, which will lead the proposed theoretical work to be inapplicable to many widely-used point processes, e.g., Hawkes process and self-correcting process, whose intensities are not continuous. Because the authors apply some complicated point process models, e.g., neural Hawkes process, and achieve encouraging performance, this assumption may be redundant or can be relaxed. In particular, I wonder if the assumption of Riemann integrable can be replaced with Lebesgue integration? Overall, I think it is nice work, but the conflict on the assumption and the experimental settings prevents me from accepting this work directly. Minors: The font size of texts in figures should be enlarged. The information of the last reference (Xu et al. 2018) is wrong. It was published at IJCAI.

Correctness: Yes.

Clarity: Yes.

Relation to Prior Work: Yes.

Reproducibility: Yes

Additional Feedback: After the rebuttal I tend to accept this paper. I am satisfied with my concerns being addressed and the addition of the new experiments to the revised version.


Review 2

Summary and Contributions: The paper proposes a novel noise-contrastive estimation for multivariate point processes. The authors evaluate their method on both synthetic and real-world datasets, and show that the proposed method takes much less wall-clock time while still achieving competitive log-likelihood.

Strengths: The idea of applying NCE to point process looks interesting, even though it is not the first time to be proposed. The research question of finding an efficient estimator is relevant to the NeurIPS community. The paper is well-written and clear.

Weaknesses: The NCE estimator for MPP looks interesting. However, the paper suffers from a number of flaws that should be better addressed. 1. The paper proposes an NCE estimator for MPP. However, this is not the first attempt to apply NCE for point processes. The INITIATOR model (Guo et al., 2018) has already attempted to do so. I believe the extension from univariate point processes to multivariate ones should not be considered as a significant contribution. 2. Whether the advantages of NCE are applicable to point processes is a question. The main benefit of NCE is to reduce the computational cost of MLE. However, the proposed method involves a sampling procedure, which is usually time-consuming. The authors also fail to consider and compare with other existing estimators for point processes that are more efficient than MLE. For example, the least-square estimator, (which has been integrated into the python library “tick” for learning point processes) even has a closed-form solution for learning linear multivariate Hawkes processes. Further, the broader category of martingale estimator, which LSE falls in, also possesses the desired properties of consistency and asymptotic normality. These commonly-used methods should also be mentioned and discussed. 3. The theoretical properties seem to be inherited from the NCE, rather than being derived from the proposed incorporation. 4. The empirical evolution is week. The paper only involves one baseline (NHP) with MLE as the underlying ground truth. More baselines should be considered, such as parametric point processes (vanilla Hawkes processes), the recurrent marked point processes, etc.

Correctness: The claims and method seem to be correct.

Clarity: The paper is well written and clear.

Relation to Prior Work: The paper missed quite a few methods that should be taken into account: the least-sequare estimator for point processes, the recurrent marked point processes, as well as many parametric point process models.

Reproducibility: Yes

Additional Feedback: Please see above.


Review 3

Summary and Contributions: This paper proposes a noise-contrastive estimation for point process which is expected to compute efficiently. The authors also prove that optimality can be achieved under mild assumptions. Empirical experimental results are used to demonstrate its efficiency and usefulness.

Strengths: They develop a new learning algorithm for point process using the idea of contrastive noise estimation. Optimality and efficiency is guaranteed through theoretical analysis.

Weaknesses: As there are already methods like work of Guo et.al which speed up the learning process of point process, addition of the comparision with those methods makes the experiment more convincing.

Correctness: The logic is sound and supported by experiments.

Clarity: The paper is easy to follow and clearly written.

Relation to Prior Work: Related works are well addressed.

Reproducibility: Yes

Additional Feedback:


Review 4

Summary and Contributions: While the authors' response addressed some of my concerns it is not enough to raise my rating. The paper describes how noise contrastive estimation can be used to train generative models of multivariate point-processes in continuous time. The authors show that training is faster, and in most cases of similar quality, than training with maximum likelihood estimation. The authors proof that under mild assumptions their method fulfils theoretical guarantees and converges to the true parameters for infinite data. They apply the method to multiple synthetic and real datasets and show how different parameters affect the outcome of the training and perform ablation studies. The authors discuss related works and give their thoughts on the broader impact of their work.

Strengths: The paper is clearly written, describes well what is needed to change NCE to work for multivariate point processes and gives enough information to fully reproduce their results. It is a good contribution and the authors provide ample theoretical grounding for their claims and evaluate their method on a range of datasets. They provide multiple ablation studies and discuss their choice of parameters in detail, especially in the supplementary material.

Weaknesses: I was missing a discussion and comparison of other ways to approximate the log likelihood, e.g. variational approximations or monte carlo estimates. It would also be interesting to see what simple baseline log likelihood models would have achieved on the data. The authors show that for some of the data using a Poisson process as q achieves very good results but not if assuming p to be a simpler model would work as well. In general the related works section could be a bit broader to touch on methods beyond NCE. While the authors compare runs of NCE with different values for parameters like C and M, it would have been more informative to show a plot of the relationship of these parameters to convergence speed directly, instead of just having multiple runs in the same likelihood plot. I think it is a good contribution but not a huge step from prior work on NCE for point processes. Given that the main advantage over training with MLE is the computational complexity it would also be nice to have shown its results on data where MLE is not feasible.

Correctness: Yes, I haven't found any incorrect statements.

Clarity: The paper is overall very well written and easy to understand. The plots are very small on the papers making some of the annotations impossible to read if printed and only readable if zoomed in closely on a computer.

Relation to Prior Work: Yes, the authors clearly distinguish from prior work and explain limitations of similar approaches and how they needed to modify the method to work for multivariate point processes in continuous time.

Reproducibility: Yes

Additional Feedback:

[Author Response · NeurIPS 2020]

We ran **new experiments** to address your concerns: you'll like the results! These results (and related discussion) could
be easily included in camera-ready, using supplementary material + the extra page that NeurIPS 2020 allows.

**New experiment A (R3, R4): comparison with Guo et al. & LSE.** We ran experiments to compare with Guo et al.
and least-squares estimation (LSE) on multivariate point processes. Figure 1 shows the learning curves of MLE (red), our
NCE (blue), Guo et al. NCE (green) and LSE (orange). Both Guo et al. and LSE converged (eventually) to much worse
   log-likelihood than our method and did so more slowly. We promise to use larger figures and fonts for the final version.

**Figure 1:** Learning curves on Synthetic-2 and BitcoinOTC datasets. ($x$-axis is truncated.) Similar patterns hold on all our other datasets, with the best curve always being NCE.

**New experiment B (R2): prediction accuracy.** Models that achieved comparable log-likelihood—whether they were
trained by MLE or NCE—achieved comparable prediction accuracies (measured by RMSE for time and Error Rate for
type). Therefore, NCE still beats MLE at converging quickly to the highest prediction accuracy.

**New experiment C (R3, R5): simple baseline models.** We checked the classical Hawkes process on our datasets,
with MLE training. Classical Hawkes performed far worse on both training and test data than the deep models in our
paper (similar to the experiments of Mei & Eisner 2017). The deep models have more flexibility to fit the data.

Now we clarify our **contributions**: new method, new theorems, sampling speedup, analysis of runtime, etc.

**Mild assumption (R2).** No, our theorems only require the intensity functions to be Riemann integrable, *not* continuous!
Indeed, in our experiments, they are typically *discontinuous* at events, though continuous between events (line 55). This
setting is Riemann integrable, as are the other widely-used point processes that R2 mentioned—so our results apply.

**New theorems (R3).** Did we merely inherit the theoretical properties from the discrete case? No, we needed non-trivial
additional work. Lemma 1 in Appendix A showed that if $\theta$ and $\theta^*$ are meaningfully different in that they predict
different intensities at time t for *some history*, then they actually do so for a *set of histories of non-zero measure*, making
this difference visible in the objective function. (Note to R2: This lemma does require Riemann integrability, not
Lebesgue integrability.) Previous work did not encounter this since they worked on non-sequential data (e.g., Gutmann
& Hyvarinen 2010 + 2012) or discrete-time sequential data (e.g., Ma & Collins 2018). We'll highlight this difference.

**New method (R3, R4).** There are two approaches to NCE—BINARY and RANKING. We chose RANKING because
we are working with conditional intensity functions. Our key idea of how to apply this to continuous time (line 111)
is new, and required new analysis. Guo et al. used the older BINARY version, which is *not* well-suited to conditional
distributions (see Ma & Collins 2018). This complicates their method since they needed to build a parametric model of
the local normalizing constant, giving them weaker theoretical guarantees (see lines 243–247) and worse performance
(Figure 1 above). So our contribution was *not* (merely) to extend to multivariate point processes as R3 implies.

**Sampling speedup + analysis of runtime (R3).** Sure, NCE requires sampling. But so does MLE (lines 73–74). We
compared them **analytically** (§3.2) and showed **experimentally** (§5) that NCE evaluates on *fewer* samples and is
practically faster (often by a factor of 5–10), in part because of our efficient method for sampling NCE noise (§3.1).

**Least-squares estimator (R3).** For classical or neural Hawkes processes, LSE is not faster than MLE—it requires
sampling to estimate an integral, just like MLE. (R3 notes that it is faster for linear Hawkes processes, but those are
extremely simple and of limited use.) In practice, LSE underperforms MLE in our experimental settings: see Figure 1.

**Other methods (R5).** We *did* use a Monte Carlo estimate for the log-likelihood. MCMC and variational methods
aren't needed in our experiments, because the complete-data likelihood is defined by the simple equation (2). It's
simple because our point process models are autoregressive; there's no need to fit a tractable lower bound (ELBO) as
in globally normalized models. It contains no difficult log-sum-exp, only an integral over a summation. This can be
estimated directly and without bias by sampling, which is what we do. (So isn't MLE always "feasible"?)

**More evaluation (R3).** Our evaluation was actually rather wide-ranging for an 8-page paper that also includes theorems.
We evaluated on 6 (dataset, model) pairs: 2 synthetic and 4 real, spanning both NHP and NDTT models. Our baselines
included both MLE and ablated versions of our full NCE method. We had so many results that many of them (figures &
discussion) went to an appendix—maybe they were missed? We established that NCE is often more efficient than MLE
and thus is worth trying in practice. We don't think this conclusion would change by testing the method on other similar
parametric point process models. An 8-page paper needn't test a method on every conceivable (dataset, model) pair.

**Other (R2, R3, R5).** We'll take other suggestions, including correcting the venue of Xu et al. (R2), using larger fonts
in figures (R2, R5), directly comparing convergence for diff. hyperparams (R5), and broader impact on society (R3).

[Meta-Review · NeurIPS 2020]

The paper derives a new estimation method for multi-variate point processes that is based on the 'ranking'-variant of NCE. The paper is borderline: two reviewers think that the difference to previous work by Gao (who use NCE to estimate point-processes) and the empirical comparison is not sufficient. Two other reviewers disagree, with one in particular arguing that the paper should be accepted. The meta-reviewer thinks that the theory in the paper is sufficiently different from Gao's work, and that the theoretical aspects of the paper are deeper and more rigorous. The results do not follow directly from previous work by Gutmann & Hyvarinen (2012) or Ma & Collins (2018). The empirical results are good and the method should be useful in practice. Moreover, the additional results provided in the rebuttal demonstrate compellingly the advantage compared to previous work. For these reasons, the meta-reviewer is in favour of accepting the paper, requiring however that (using the additional space available for camera-ready papers): - empirical comparisons to Gao's work and the least-square estimator are added to the examples considered in the paper (much like the rebuttal, but with a detailed description of the setup and tuning parameters used) - the discussion of related work, in particular the differences to Gao's work, are expanded - the figures are presented such that all labels are legible. (Of course, the changes promised in the rebuttal need to be implemented and the reviewers' comments taken into account when revising the paper) Additional comments: - The sentence "Our method is a version of noise-contrastive estimation (NCE), which was originally developed for softmax distributions such as language models." is wrong. NCE is a general estimation principle that was developed for unnormalised (energy-based) models. First applications were in natural image statistics. - Appendix C2 discusses the idea of using as noise distribution a model previously learned with NCE. Please note and acknowledge that this has already been considered in the original NCE paper in 2010.